# SID: MULTI-LLM DEBATE DRIVEN BY SELF SIGNALS

## ABSTRACT

Large Language Models (LLMs) have exhibited impressive capabilities across diverse application domains. Recent work has explored Multi-LLM Agent Debate (MAD) as a way to enhance performance by enabling multiple LLMs to discuss and refine responses iteratively. Nevertheless, existing MAD methods predominantly focus on utilizing external structures, such as debate graphs, using LLM-as-a-Judge, while neglecting the application of self signals, such as token logits and attention, that arise during generation. This omission leads to redundant computation and potential performance degradation. In this paper, we shift the focus to the self signals of multi-LLM debate and introduce a Self-Signals Driven Multi-LLM Debate (SID), which leverages two types of self-signals: model-level confidence and token-level semantic focus, to adaptively guide the debate process. Our approach enables high-confidence agents to exit early at the model level and compress the redundant debate contents based on the attention mechanism. We evaluate our method on various LLMs and Multimodal LLMs across multiple challenging benchmarks. Experimental results demonstrate that our method not only outperforms existing MAD techniques in accuracy but also reduces token consumption, highlighting the effectiveness of utilizing self signals in enhancing both the performance and efficiency of multi-agent debate systems.

## 1 INTRODUCTION

Large Language Models (LLMs) have demonstrated remarkable capabilities (Brown et al., 2020; Kojima et al., 2023) across a wide range of domains, including science, technology, engineering, mathematics (STEM) questions (Hendrycks et al., 2021; Wang et al., 2024), and complex reasoning tasks (Rein et al., 2023). The emergence of Multimodal LLMs (MLLMs) further extends the potential to the visual input domain (Lu et al., 2022; Li et al., 2023; Liu et al., 2024a). However, current models still suffer from inherent limitations such as inaccuracies and hallucinations.

Multi agent debate (MAD) offers an orthogonal approach to enhancing model performance, in which multiple agents iteratively discuss and refine their answers accordingly (Du et al., 2024; Liu et al., 2024c; Sun et al., 2025). However, a challenge arises from the prevalence of redundant content and repeated consensus points during debate, which not only waste computational resources but also introduce informational noise, potentially impairing the agents' final judgments (Du et al., 2024; Li et al., 2024b). Moreover, this iterative discussion paradigm incurs substantial token overhead, which becomes increasingly incongruent with the growing capabilities of modern foundation models (OpenAI et al., 2025). This inherent contradiction between performance gains and token consumption cost presents a central dilemma in contemporary MAD research.

To alleviate this problem, several optimization strategies have been proposed. Broadly, these methods typically fall into two categories: (i) structural optimization, such as adopting various prompting skills (Liu et al., 2024c), reducing communication via sparse debate graphs or clustering agents into local debate groups (Liu et al., 2024b); and (ii) history management, including summarization of prior discussions or introducing agent self-generated confident score (Sun et al., 2025). Whereas these approaches improve the efficiency of information flow in *external* ways (*i.e.,* restructuring agent communication or using LLM-as-a-judge to interpret history), they often suffer from secondary errors such as hallucinations in judges or summaries as evident in (Xiong et al., 2023; Zhang et al., 2024; Tian et al., 2025)). This limitation motivates us to think: *can we avoid relying on error-prone external mechanisms, and instead leverage more reliable self signals from each agent's generative process to prevent unnecessary and potentially wasteful debate?*

Motivated by the above, in this work, we present a framework that leverages self signals available during LLM inference to improve debate efficiency and performance. In this framework, two types of signals: *model-level confidence* and *token-level semantic focus*, are extracted and used to provide complementary guidance for distinguishing essential information from redundancy, thereby enhancing overall debate quality and efficiency. The model-level confidence, estimated from the probability distribution over the initially generated answer, quantifies how certain the model is about its response. We leverage this signal to design an ***early-exit mechanism*** that avoids invoking debate when the model is already sufficiently confident, thereby reducing potential noise and redundancy. The token-level semantic focus, derived from attention patterns conditioned on disagreement-oriented prompts, identifies spans in the debate content that the model considers semantically relevant to the disagreement among different agents. We extract and reconstruct these high-attention spans to form a more compact context, thereby introducing a ***compression mechanism*** that preserves critical points of contention while significantly reducing token overhead.

By integrating these two mechanisms, each leveraging a different level of self signal, we propose a unified Self **Si**gnal Driven **D**ebate framework (SID) to enhance LLM performance. This framework enables early exit for confident agents and extracts focused context for the remaining ones, dynamically adapting the debate process based on the model's own epistemic signals. We evaluate our method across multiple LLMs and MLLMs on diverse benchmarks, including MMLUpro, Math, GPQA, ScienceQA, and MMstar. SID consistently outperforms existing MAD approaches in most scenarios, while also achieving up to a 40% reduction in token consumption. These results demonstrate the strong effectiveness of our approach and highlight the significant potential of leveraging internal belief signals in multi-agent systems to jointly optimize performance and efficiency. Our key contributions can be summarized as follows:

- We present **SID**, a multi-agent debate framework that leverages self signals from the LLM generation process to enhance agent debate.
- We instantiate two types of LLM self signals: model-level confidence and token-level semantic focus, and leverage them to design an early-exit and a compression mechanism, respectively, effectively reducing redundancy and enhancing debate performance.
- Integrating the two proposed mechanisms, we construct an effective and efficient debate framework, SID. Experiments across multiple benchmarks, on both LLMs and MLLMs, demonstrate the significant advantages of SID over existing methods.

## 2 RELATED WORK

**Reasoning Augmentation** To enhance the reasoning capabilities of LLMs, researchers have explored various techniques. Early work primarily focused on guiding the model through step-by-step reasoning through Chain-of-Thought (CoT) prompts (Wei et al., 2023) or generating multiple reasoning paths (self-refinement) and voting for the optimal solution through self-consistency or using multi-round self-reflection (Zhang et al., 2024; Yao et al., 2023). Additionally, subsequent research has found that the model's self-correction capabilities are limited, leading to stagnation in reasoning quality (Zhang et al., 2024). This has partially motivated the rise of multi-agent collaborative paradigms, particularly multi-agent debate (MAD) (Du et al., 2024), which introduces external perspectives and dynamic feedback among agents to overcome the limits of self-reflection. Our work differs from these studies in that they focus primarily on improving reasoning ability through context prompts, whereas we propose to use self-signals from a model to optimize the context prompt at the token level, thus improving the effectiveness of performance and token ratio.

**Uncertainty Analysis** Uncertainty in LLMs is typically categorized into aleatoric (data-related) and epistemic (model-related) uncertainty (Kiureghian & Ditlevsen, 2009; Gawlikowski et al., 2023; Hu et al., 2023; Ye et al., 2025). Given the structured nature of current tasks (e.g., QA, math, science), recent works have focused on quantifying epistemic uncertainty. Mainstream approaches include: (i) probability-based metrics, such as token-level entropy or negative log-likelihood (Tu et al., 2025) on the level of the attention layer (Schuster et al., 2022; Laaouach, 2025; Corallo & Papotti, 2024) and reasoning chain (Yang et al., 2025); (ii) ensemble-based methods, e.g., Monte Carlo sampling (Metropolis et al., 1953; Hastings, 1970) and Bayesian methods (Kwon et al., 2020); and (iii) verbalization-based techniques that prompt the model to self-report confidence (Tian et al., 2023).

Among these, probability-based methods are especially attractive due to their seamless integration with the generation process, without requiring multiple generations, which incur significant token overhead. Recent work, such as ReConcile and DebUnc (Xiong et al., 2023; Chen et al., 2024a; Yoffe et al., 2025; Kirchhof et al., 2025) further explores agent-level uncertainty in interactive settings, emphasizing the role of uncertainty as a confident signal for learning and output control. Our method differs from theirs in model-level adaptive debate scheduling and token-level compression instead of passing the full history. Our method aligns with the uncertainty in the interactive setting, leveraging self-signals as dynamic indicators of agent-level uncertainty to control agent participation during debates.

**Multi LLM Debate Systems** Previous multi-LLM debate employs a role-playing setup (Liang et al., 2024), which has been demonstrated strengths in collaborative tasks. Subsequent research has shown that it is less suited for certain types of problem-solving scenarios. Multi Agent Debate (MAD) (Du et al., 2024) introduces external perspectives to enrich the system's reasoning capabilities. DMAD (Liu et al., 2024c) proposes specialized prompt strategies to diversify agent behavior. S2-MAD (Zeng et al., 2025) introduces a selective sparsity mechanism, allowing agents to selectively participate based on internal cues. CortexDebate (Sun et al., 2025) constructs a dynamic sparse debate graph by letting agents serve as self-judges and output confidence scores. These works focus on improving performance via external states (*e.g.,* optimizing structures, using LLM-as-a-Judge). Compared to previous approaches (Zhang et al., 2025), our method can be orthogonal and complementary, which provides a new angle by integrating self-signals into the debate process instead of optimizing communication structures.

## 3 PRELIMINARIES

We first introduce the naive multi agent debate paradigm in this section. Let $\mathcal{V}$ denote the vocabulary and $\mathrm{Tok}$ the tokenizer. Given a query (e.g., natural language, image, and text) $Q$, $\mathbf{x} = \mathrm{Tok}(Q)$ is the tokenized prompt. An casual LLM $M$ produces a response sequence $\mathbf{y} = (y_1, \ldots, y_m)$ with per-step logits $\boldsymbol{\ell}_t \in \mathbb{R}^{|\mathcal{V}|}$ and probabilities $\boldsymbol{\pi}_t = \mathrm{softmax}(\boldsymbol{\ell}_t)$. A debate involves $n$ agents $\mathcal{A} = \{1, \ldots, n\}$ over rounds $t = 0, 1, \ldots, T$. Let $\mathbf{y}_t^{(j)}$ be agent $j$'s response at round $t$; round 1 is the initial answering round without debate context. The per-round input to agent $j$ at round $t+1$ concatenates the query, its own last response, and other agents' last responses:

$$\mathbf{x}_{t+1}^{(j)} = \mathrm{Tok}\Big(Q \parallel \mathbf{y}_t^{(j)} \parallel (Concat_{k \neq j}\mathbf{y}_t^{(k)})\Big). \tag{1}$$

Here, both $\parallel$ and $Concat$ represent concatenation between prompt groups.

## 4 METHOD

The above naive framework suffers from several issues, such as excessive redundancy and low efficiency. To address these challenges, as shown in Algorithm 1 and Figure 1, we propose Self Signal Driven Debate (SID), a framework that leverages internal confidence signals readily available during inference to adaptively guide the multi-LLM debate process. Specifically, SID utilizes two types of self signals from the LLM: *model-level confidence* and *token-level semantic focus* (see examples in Figure 3 and the Appendix F). *Model-level confidence*, derived from the token-wise output probability distribution (logits), reflects how confident an agent is in its initial answer. We leverage this signal in a newly designed early-exit mechanism to enhance debate efficiency. *Token-level semantic focus*, extracted from the self-attention maps conditioned on disagreement-oriented prompts, captures regions of high variability and knowledge density throughout the debate. This signal is incorporated into a compression mechanism to alleviate token redundancy. In the following two sections, we introduce these two components in detail.

### 4.1 EARLY-EXIT WITH MODEL-LEVEL CONFIDENCE

We first introduce an early-exit mechanism to mitigate redundant debate, motivated by the intuition that cross-LLM discussion is more necessary when a single model lacks confidence in its response.

**Algorithm 1** Self-Signal Driven Debate (SID)

**Require:** Query $Q$; LLM agents $\{M_j\}_{j=1}^m$; rounds $N$; confidence threshold $\theta$; top-$p$ ratio $\rho$;
1: $Y_1 \leftarrow M_1(Q)$
2: $u \leftarrow \phi_U(U(Y_1))$       ▷ Model-level Confidence(Sec. 4.1)
3: **if** $u \leq \theta$ **then**
4:     **return** $Y_1$
5: **end if**                      ▷ early exit
6: $Y_1 \leftarrow \{y_1\};\quad y_1^{(j)} \leftarrow M_j(Q)$ for $j=2..m$
7: **for** $t = 2$ to $N$ **do**
8:     **for** $j = 1$ to $m$ **do**
9:        $X_t^{(j)} \leftarrow \left( Q \,\|\, y_{t-1}^{(j)} \,\|\, \texttt{[PROMPT]} \,\|\, \text{Concat}_{k \neq j}(Y_{t-1}^k) \right)$
10:       $A_t^{(j)} \leftarrow \text{FORWARDATTENTION}(M_j, X_t^{(j)})$    ▷ forward only
11:       $\widehat{\mathcal{C}}_t^{(j)} \leftarrow \text{TopP}(A_t^{(j)}, \rho)$
12:       $\mathcal{S}_t^{(j)} \leftarrow \text{SemanticPreserve}(\widehat{\mathcal{C}}_t^{(j)})$     ▷ Sec. 4.2
13:       $y_t^{(j)} \leftarrow M_j\left( Q \,\|\, y_{t-1}^{(j)} \,\|\, \mathcal{S}_t^{(j)} \right)$    ▷ generate with compressed context
14:     **end for**
15:     $Y_t \leftarrow \{y_t^{(j)}\}_{j=1}^m$
16: **end for**
17: **return** $Y_N$

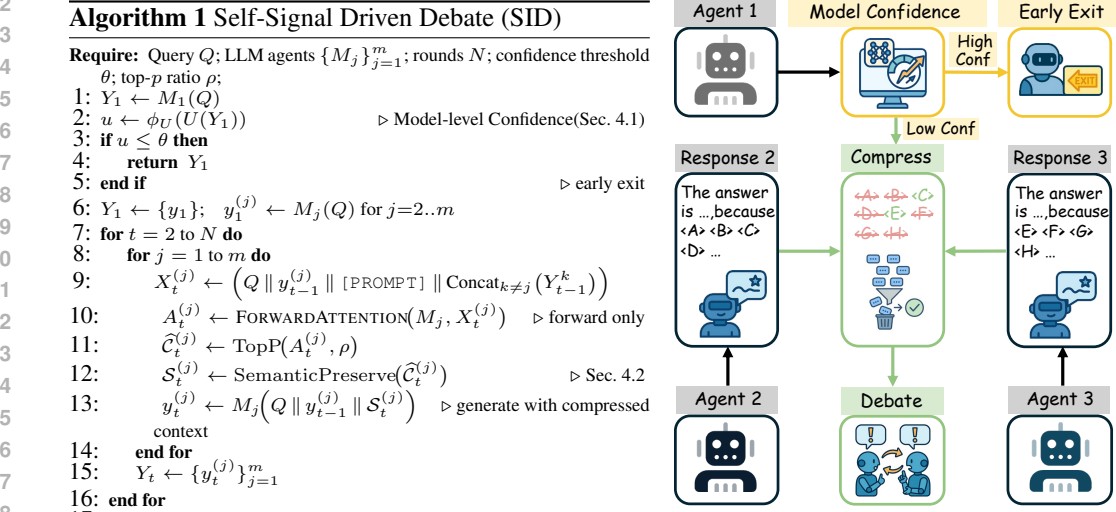

Figure 1: Overall framework of SID

The key to this mechanism lies in extracting an effective confidence score. Intuitively, the more peaked the model's output distribution over the vocabulary (i.e., lower entropy), the more confident it is in its prediction. Based on this motivation and following conventional methods (Tu et al., 2025), we adopt two token-wise uncertainty metrics: entropy $\text{Ent}(\boldsymbol{\pi}_t) = -\sum_{v \in \mathcal{V}} \pi_{t,v} \log \pi_{t,v}$ and negative log-likelihood $\text{NLL}_t = -\log \pi_{t,y_t}$, to estimate the confidence for a generated answer $\mathbf{y}$. To aggregate token-level metrics into a sequence-level confidence score, we explore four aggregation strategies: (1) averaging over tokens (average), (2) taking the maximum value (max), (3) using the first token's value (first), and (4) using the penultimate token's value (penultimate). This yields eight confidence measures (four for entropy and four for NLL). Each variant captures different facets of uncertainty: e.g., max focuses on worst-case ambiguity, while penultimate emphasizes late-stage uncertainty often aligned with final reasoning steps in autoregressive generation. We concatenate these eight measures to form a vector $U(\mathbf{y})$, which we empirically find to be statistically significant in distinguishing incorrect answers (see Figure 2 for details).

After obtaining confidence scores, the next challenge lies in leveraging them to effectively guide model-level early exits during multi-LLM debates. To this end, we propose two different drop-in strategies that convert the agent's confidence vector into a binary decision boundary:

**Vocabulary-Adaptive Threshold**    We first tried a straightforward method by directly setting a fixed threshold on the confidence metrics across different types of models. However, this naive strategy yielded suboptimal performance, likely due to the inherent dependency of entropy and NLL magnitudes on the vocabulary size $|\mathcal{V}|$ of the underlying LLM. For example, under a uniform generation assumption, both entropy and NLL equal $\log |\mathcal{V}|$. Thus, larger vocabularies naturally induce higher values, while using a single threshold across models leads to unfairness and unreliability. Based on this motivation, we propose a vocabulary-adaptive threshold as follows:

$$\theta(V) = \alpha \log |\mathcal{V}|, \quad \text{and decide Terminate iff } \phi_U(U(\mathbf{y})) \leq \theta(V), \tag{2}$$

where $\alpha > 0$ is a hyper-parameter, $\phi_U$ is an operator to filter noisy metrics $U(\mathbf{y})$. This strategy ensures fair and consistent confidence evaluation across models with varying vocabulary sizes.

**Calibrated Confidence**    While the aforementioned method provides a simple and robust solution, it relies on a uniformity assumption over token distributions that may not hold in practice. To capture more nuanced confidence signals, we introduce an alternative method, using a lightweight nonlinear classifier $C : \mathbb{R}^d \to [0, 1]$ trained over a small held-out set. This model takes the confidence vector as input and outputs a scalar confidence score, calibrated against correctness labels:

$$\text{Terminate iff } C(U(\mathbf{y})) \geq \tau_c, \quad \tau_c \in (0, 1). \tag{3}$$

**Confidence-Guided Early Exit**   In practice, we adopt the vocabulary-adaptive threshold for gating due to its sufficiently strong performance and training-free simplicity. Specifically, gating is applied in the first round: if an agent's confidence reaches a high value, it is terminated early, signalling that the system is already sufficiently confident in its answer (see Appendix Figure 17–20 for examples). Conversely, if the model exhibits low confidence, this suggests that the question is sufficiently challenging and unlikely to be resolved without additional reasoning, thereby motivating the initiation of a multi-agent debate with the input described in Eq.1.

## 4.2 Adaptive Compression With Token-Level Semantic Focus

In addition to model-level confidence, we further exploit another self signal from the LLM: token-level semantic focus, to improve debate efficiency. As a debate progresses, the accumulated context from multiple agents often becomes repetitive and redundant. We observe that this increasing redundancy can dilute the signal-to-noise ratio, potentially degrading the effectiveness and efficiency of the debate process. A common approach to mitigate this is to use LLM-as-a-judge to summarize past exchanges. However, this method is limited by the summarization capabilities of the base models, which can be prone to hallucination or information loss (Li et al., 2024a). To address this limitation, we instead leverage attention, an intrinsic mechanism of transformer-based models that naturally reflects the model's focus and the debate's salient region, as an internal signal to implement a token compression framework.

**Prompt-conditioned Attention Extraction**   Given the query $Q$, the agent $j$'s previous answer $\mathbf{y}_t^{(j)}$, and other agents' responses $\{\mathbf{y}_t^{(k)} : k \neq j\}$, we construct a concatenated input:

$$\mathbf{x}_{t+1}^{(j)} = \mathrm{Tok}\left(Q \parallel \mathbf{y}_t^{(j)} \parallel \texttt{[PROMPT]} \parallel \left(Concat_{k \neq j}\mathbf{y}_t^{(k)}\right)\right), \tag{4}$$

where `[PROMPT]` is a task instruction. Here, we use the prompt: `"Identify the key points where they disagree with your own reasoning. Concentrate on those disagreements and decide which line of reasoning is better."`, motivated by prior work demonstrating the benefits of identifying disagreements among agents (Du et al., 2024; Zeng et al., 2025). This prompt directs the model's attention toward segments of the debate that involve semantic conflict, thereby enhancing its focus on critical reasoning divergences. We then define $\mathcal{Q}$ as the set of token positions within the injected prompt, $\mathcal{C}$ as the positions corresponding to other agents' responses (*i.e.,* $Concat_{k \neq j}\mathbf{y}_t^{(k)}$), and $A^{(l,h)} \in [0,1]^{L \times L}$ as the attention score at layer $l$ and head $h$. For $c \in \mathcal{C}$, a prompt-conditioned semantic focus score is computed by:

$$s(c) = \max_{l,h} \max_{q \in \mathcal{Q}} A_{q,c}^{(l,h)}, \tag{5}$$

which represents the maximum attention weight from any prompt token to $c$ across all heads and layers, capturing the extent to which $c$ is considered relevant to the disagreement-focused instruction.

**Compression with Semantic Preservation**   While token-level attention scores $s(c)$ enable fine-grained identification of salient contents, directly selecting individual tokens may result in fragmented phrases or broken sentence structures. Such fragments hinder the model's ability to interpret the compressed input coherently. To address this, we apply a semantic preservation heuristic that extends high-attention tokens to complete sub-sentential units. Concretely, we first select the top-$p$ fraction of context tokens, forming $\widehat{\mathcal{C}} = \mathrm{Top}_p\{(c, s(c))\}_{c \in \mathcal{C}}$. Using the tokenizer's offset map $\Psi : c \mapsto [T_a(c), T_b(c)]$, we merge overlapping spans and then expand to sentence boundaries to preserve semantics information. To ensure semantic completeness, we then expand each segment to align with syntactic boundaries, such as commas, periods, or coordinating conjunctions. We denote this process as the $\mathrm{SemanticPreserve}$ operation (see Appendix C.1, Figure 6 for implementation details), which produces a minimal set of semantically coherent text spans as follows:

$$\mathcal{S} = \mathrm{SemanticPreserve}\left(\mathrm{Merge}\left(\{\Psi(c) : c \in \widehat{\mathcal{C}}\}\right)\right). \tag{6}$$

We denote the compressed textual summary for agent $j$ as $\text{Text}(\mathcal{S})$, and the next-round input in the debate process (Eq.1) becomes:

$$\widehat{\mathbf{x}}_{t+1}^{(j)} = \text{Tok}\Big( Q \parallel \mathbf{y}_t^{(j)} \parallel \text{Text}(\mathcal{S}) \Big). \tag{7}$$

In practice, replacing full histories by $\text{Text}(\mathcal{S})$ yields substantial token compression while preserving points of disagreement.

### 4.3 OVERALL METHOD

Based on the aforementioned early-exit method with model-level confidence (Sec. 4.1) and adaptive compression mechanism with token-level semantic focus (Sec. 4.2), we then present the overall SID framework. As shown in Figure 1, after initial generation, each agent assesses its confidence using token-level uncertainty metrics derived from output logits. If the agent is sufficiently confident, it exits the debate early, avoiding unnecessary interaction. For less confident cases, the debate proceeds with a compression mechanism guided by the model's own attention dynamics. A disagreement-oriented prompt steers the attention toward semantically relevant spans in other agents' responses. These spans are then selected and reconstructed into a concise context for the next round, preserving key points of contention. By coupling generation-time uncertainty with attention-driven semantic focus, SID adapts the debate trajectory according to each agent's internal belief state, achieving both high efficiency and robustness without additional training. Readers could refer to Algorithm 1 for a more detailed illustration of the overall implementation.

## 5 EXPERIMENTS

### 5.1 EXPERIMENT SETUP

**Tasks and Benchmarks.** Results on both LLM and MLLM tasks are presented. For LLM tasks, we evaluate our method on MMLUpro (Wang et al., 2024), and Math (Hendrycks et al., 2021) datasets, as they represent a wide range of problem-solving tasks in different domains. For MLLM tasks, we evaluate on ScienceQA (Lu et al., 2022) and MMStar (Chen et al., 2024b) datasets. Taken together, these four benchmarks span (i) text-only vs. multimodal inputs, (ii) factual, analytical and symbolic reasoning, and (iii) both LLM and MLLM settings, forming a compact yet diverse testbed for multi-agent debate methods. More expanded experiments can refer to Appendix D . In consistent with previous methods, we randomly sample 100 questions from each dataset for evaluation. For the ScienceQA dataset, we utilize the lecture and hint as additional text information following (Liu et al., 2024c). For all other datasets, we adopt a zero-shot prompt setting by default.

**Models.** To ensure representative coverage of different foundation models, we evaluate both general-purpose and reasoning-oriented models. For LLM tasks, we test on LLaMA-3.1-Instruct-8B (LLaMA3.1-8B) Grattafiori et al. (2024) and the recently released GPT-OSS-20B OpenAI et al. (2025). For MLLM tasks, we evaluate LLaVA-v1.6-Vicuna-13B (Hugging Face version, LLaVA1.6-13B) and the GLM4.1V-Thinking (GLM4.1V) reasoning model (Team et al., 2025).

**Implementation Details** We follow the setup of prior work (Du et al., 2024; Liu et al., 2024c) to ensure fair comparison, using $n = 3$ agents and $N = 2$ debate rounds across all SID, MAD, and DMAD settings. The number of self-consistency samples is set to 3. Additionally, we incorporate step-back prompting (Zheng et al., 2024) and self-contrast (Zhang et al., 2024) as reasoning augmentation methods in complement to IO (directly output) and COT methods. For model-level confidence, we set the NLL-max threshold $\alpha$ to 1.0 for reasoning-oriented models, 0.5 for general-purpose models, and 0.25 for MLLMs. To mitigate the impact of attention sinks and special tokens on specific token logits (Xiao et al., 2024), we empirically set $\phi(U)$ as the maximum of NLL and entropy, and exclude certain position metrics when computing model-level confidence. The confidence calibration method is trained on a held-out set of 50 samples with $\tau_c$ as 0.9. More implementation details are presented in Appendix C.

**Evaluation Metrics** For the Math dataset (Hendrycks et al., 2021), we adopt the official exact match metric to evaluate agent responses. For all other question-answering datasets, which consist of multiple-choice questions, we use accuracy as the evaluation metric.

Table 1: Performance comparison across different LLMs for various datasets (Math subsets and MMLUpro). SID-v and SID-c denote our method using the vocabulary-adaptive threshold and calibrated confidence, respectively, to implement the early-exit mechanism. (see Sec.4.1 for details)

| Model | Method | Alg. | C&P | Geo. | Int.A. | Num | Pre.A | Pre.C. | MMLUpro | Avg |
|-------|--------|------|-----|------|--------|-----|-------|--------|---------|-----|
| LLaMA3.1-8B | COT | 61 | 38 | 34 | 14 | 37 | 54 | 28 | 39 | 38.13 |
| | IO | 65 | 37 | 35 | 15 | **46** | 59 | 28 | 25 | 38.75 |
| | SBP (Zheng et al., 2024) | 46 | 28 | 21 | 12 | 33 | 46 | 24 | 15 | 28.13 |
| | Self-Consistency (Wang et al., 2023) | 58 | 25 | 32 | 12 | 40 | 55 | 25 | 45 | 36.50 |
| | Self-Contrast (Zhang et al., 2024) | 54 | 36 | 27 | 11 | 31 | 53 | 27 | 36 | 34.38 |
| | MAD (Du et al., 2024) | 61 | 36 | 36 | 16 | 37 | 60 | 29 | 41 | 39.50 |
| | DMAD (Liu et al., 2024c) | 55 | 36 | 32 | 13 | 36 | 58 | 26 | 39 | 36.88 |
| | SID-v | **67** | **43** | **40** | 18 | 41 | 64 | 31 | **47** | 43.88 |
| | SID-c | **67** | **43** | 39 | **20** | 41 | **65** | 30 | **47** | **44.00** |
| GPT-OSS-20B | COT | 85 | 81 | 56 | 36 | 70 | 84 | 44 | 61 | 64.63 |
| | IO | 85 | 81 | 60 | 40 | 74 | 87 | 42 | 64 | 66.63 |
| | SBP (Zheng et al., 2024) | 65 | 64 | 44 | 37 | 16 | 73 | 11 | 26 | 42.00 |
| | Self-Consistency (Wang et al., 2023) | 75 | 67 | 44 | 31 | 70 | 79 | 23 | 69 | 57.25 |
| | Self-Contrast (Zhang et al., 2024) | 84 | 75 | 65 | 36 | 67 | 88 | 35 | 65 | 64.38 |
| | DMAD (Liu et al., 2024c) | 91 | 90 | 73 | 51 | 66 | 89 | 47 | 65 | 71.50 |
| | SID-v | **94** | **92** | 79 | **65** | **87** | **91** | **62** | **71** | **80.13** |
| | SID-c | **94** | **92** | **80** | 62 | **87** | **91** | 61 | 70 | 79.63 |

## 5.2 MAIN RESULTS

**Overall Performance** Table 1 and Table 2 respectively present the overall performance across LLMs (including LLaMA3.1-8B and GPT-OSS-20B) and MLLMs (including LLaVA1.6-13B model and GLM4.1V) in different datasets. Our SID consistently achieves the best performance in most scenarios, demonstrating its strong effectiveness. Additionally, we observe that MAD methods outperform reasoning augmentation baselines such as self-consistency, which aligns with findings reported in (Liu et al., 2024c). Another notable observation is that both the vocabulary-adaptive threshold (SID-v) and calibrated confidence (SID-c) yield very similar performance when implementing the early-exit mechanism described in Sec.4.2. This suggests that the simple thresholding strategy can already approximate the learned decision boundary well. Given its training-free nature and practical effectiveness, we recommend SID-v as the preferred choice in real-world applications.

**Accuracy and Efficiency** Figure 2(a) compares the performance and token efficiency of our SID framework against the baseline MAD method, reporting metrics of both the accuracy and the token consumption ratio. The token ratio is computed relative to the MAD setting (i.e., MAD has a token ratio of 1). Results show that SID achieves up to a 30% reduction in token usage on science and reasoning datasets, while also attaining higher accuracy, demonstrating its significantly better efficiency and effectiveness. Note that on thinking models such as GPT-OSS and GLM4.1V, our method exhibits more significant token reduction, as their reasoning processes are inherently less amenable to token-level compression (see Figure 21,22 for examples). We also compare the *actual running times* in Figure 5 of the Appendix, where SID demonstrates substantially lower inference time, further underscoring its efficiency advantages. Additionally, Figure 2(b) presents accuracy curves across different debate rounds. SID consistently improves with additional rounds, highlighting its strong scalability under extended deliberation.

**Statistical Significance Analysis** The statistical significance of our model-level confidence metric is illustrated in Figure 2(c) and Figure 7–16, where results for both the LLM (GPT-OSS-20B) and MLLM (LLaVA1.6-13B) are presented. In the figure, C and W denote correct and incorrect responses, respectively. Across two tasks of varying difficulty: GPQA and MMLUpro, our SID maintains a consistent confidence threshold within the correct group for the same model (e.g., NLL max $\approx 7.5$), highlighting the stability and robustness of our model-level confidence signal.

## 5.3 ABLATION AND ANALYSIS

**Ablation of Key Components** Using the LLaMA3.1–8B model and the MMLUpro dataset, we conduct a comprehensive ablation study to evaluate the key design components of our framework. As shown in Table 3, the baseline MAD setup yields suboptimal performance. In contrast, incorporating our proposed early-exit mechanism based on model-level confidence (Section 4.1) and the compression mechanism guided by token-level semantic focus (Section 4.2) leads to substantial im-

Table 2: Performance on Sci.QA and MMStar based on MLLMs LLaVA1.6-13B and GLM4.1V.

| Model | Method | Sci.QA | MMStar | Model | Method | Sci.QA | MMStar |
|-------|--------|--------|--------|-------|--------|--------|--------|
| LLaVA1.6-13B | CoT | 63 | 11 | GLM4.1V | CoT | 83 | 29 |
| | IO | 62 | 9 | | IO | 83 | 32 |
| | Self-Consis | 63 | 11 | | Self-Consis | 84 | 29 |
| | MAD | 65 | 12 | | MAD | 90 | 47 |
| | SID-v | **65** | **14** | | SID-v | **91** | **54** |
| | SID-c | **65** | **14** | | SID-c | **91** | **54** |

Table 3: Ablation Study of SID on MMLUpro based on LLaMA3.1-8B.

| Method | Accuracy | Token Ratio |
|--------|----------|-------------|
| Baseline Single-round CoT | 37.67 | 0.17 |
| Baseline MAD | 39.50 | 1.00 |
| Baseline MAD + Compression | 41.67 | 0.73 |
| Baseline MAD + Compression + Early Exit | 46.83 | 0.53 |
| SID w/o Semantic Preservation | 34.50 | 0.46 |
| SID w/o Early Exit w/ Token-level Summary | 39.50 | 0.68 |
| SID | 46.83 | 0.53 |

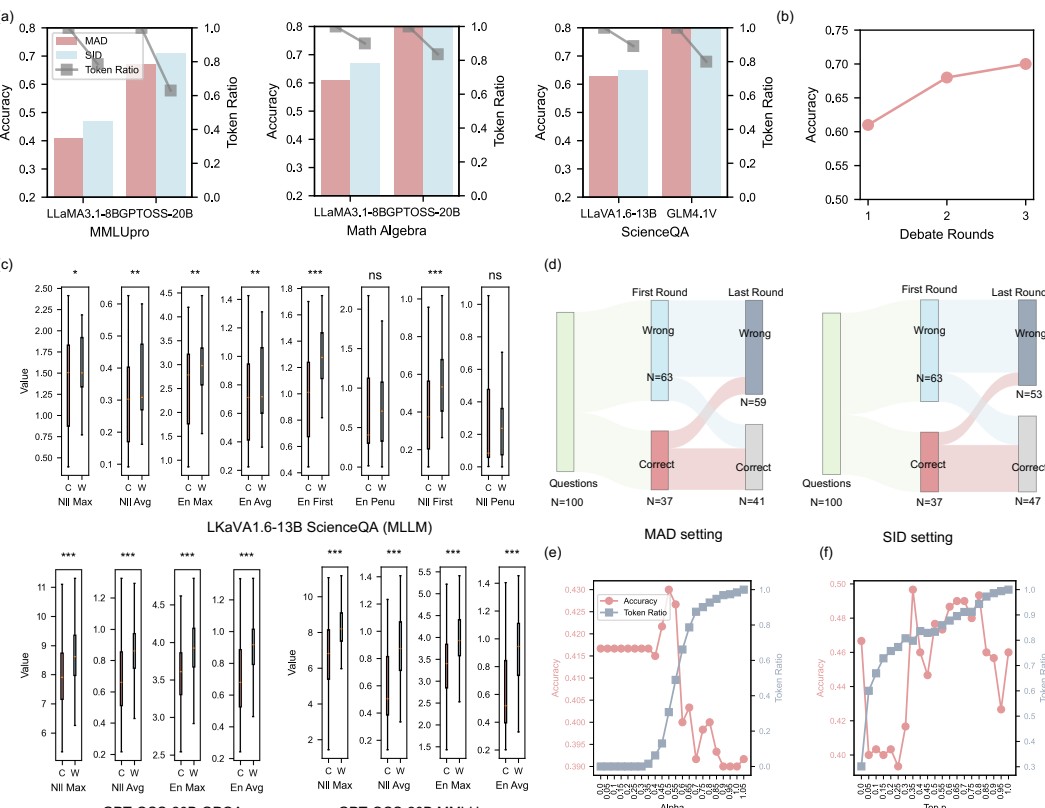

Figure 2: (a) Accuracy and token ratio comparison across strategies in MAD vs SID. (b) Performance with more debate rounds in LLM and MLLM. (c) Significance tests on model-level confidence signals. C means the correct group, and W means the wrong group. Statistical significance is indicated as follows: $p < 0.05(*)$, $p < 0.01(**)$, and $p < 0.001(***)$: (d) Answer correction flow in the MAD vs SID setting. (e) Ablation of the early-exit confidence threshold top-$p$ and (f) the semantic-preservation ratio $\alpha$ on accuracy and token ratio.

provements in both effectiveness and token efficiency. We further evaluate semantic preservation in our compression mechanism that helps enforce the completion of sub-sentential units when extracting semantic focus information. The significant performance degradation observed when this component is excluded highlights its importance and effectiveness.

Additionally, when we further enable the model-level early-exit gate on top of compression (MAD + Compression + Early Exit, equivalent to SID under this configuration), we observe a substantial additional gain: accuracy increases to 46.83% while the token ratio is further reduced to 0.53. This indicates that early exit is not merely a cost-cutting heuristic: by allowing high-confidence correct agents to stop debating early, it prevents over-debate that can otherwise corrupt correct answers, while still allocating more tokens to genuinely ambiguous cases. We also include a prompt-based self-summary variant (SID w/o Early Exit w/ Token-level Summary) as a representative LLM-as-a-judge style baseline. In this setting, after each debate round, the model is asked, via an explicit summary prompt, to first identify the key points of disagreement in the debate history and then

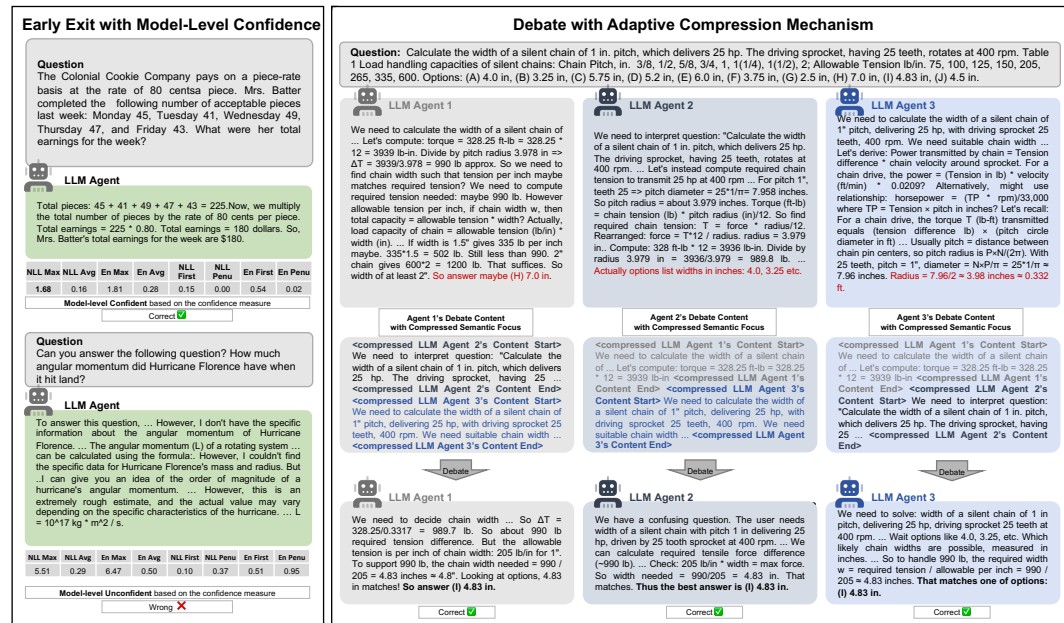

Figure 3: Case study of SID's debate process. (Left) On MMLUpro, SID exits early for a simple arithmetic question with high confidence but fails on a complex physics question with low confidence. (Right) Three agents initially err but converge to the correct answer through debate guided by token-level semantic focus from adaptively compressed content.

produce a concise summary, which is used as the context for the next round instead of our attention-based compressed history. Empirically, this self-summarization approach leads to a 7.3% absolute accuracy drop and about a 15% increase in token usage compared to our full SID configuration. As shown in Table 3, the superior performance of our SID method validates the design choices of our framework and highlight the contribution of each component to the overall performance.

**Ablation of Vocabulary Adaptive Threshold** $\alpha$  We further conduct an ablation study on the vocabulary adaptive threshold $\alpha$ and early exit ratio based on the LLaMA3.1-8B. The results are presented in Figure 2 (e). Small $\alpha$ means all questions are unconfident, thus the exit ratio is 0, equivalent to traditional MAD, whereas large $\alpha$ means all questions are confident, thus the system stops at the first round, equivalent to only one LLM model. Our results show that $\alpha = 0.5$ is an optimal value for this LLaMA3.1-8B model.

**Ablation of Semantic Preservation Ratio** $P$  In our semantic preservation framework, we select the top-$p$ fraction of context tokens for further processing. The ablation study results for varying $p$ are shown in Figure 2 (f). We observe that selecting the top tokens with $p$ around 0.35 or 0.4 yields the best performance. It is interesting to find that when $p > 0$ but very small, performance can degrade compared to the case where no additional context is included. Conversely, when $p$ is too large, which means retaining a broader range of content, including potentially redundant agreement, the performance also drops. These findings suggest that both incomplete and overly redundant context can negatively impact multi-LLM debate effectiveness.

## 5.4 VISUALIZATION RESULTS

To illustrate the mechanisms of our framework more intuitively, we present the visualizations of SID's workflow in Figure 3. The left branch showcases the early-exit mechanism on a real-world economics question. After generating an answer, the model is assessed as highly confident (e.g., NLL Max = 1.68) by the model-level confidence module and exits early with a correct prediction. In contrast, for a more complex physics question, the model is flagged as low confidence (e.g., NLL Max = 5.51), thus prompting further debate. The right branch illustrates the debate process guided by our adaptive compression mechanism. When facing a challenging physics problem, all

three agents initially fail. However, by engaging in a debate using token-level compressed content driven by semantic focus, the agents collaboratively refine their reasoning and successfully converge on the correct answer. More case studies can be found in Appendix F, Figure 22. Furthermore, Figure 2(d) compares the corrections made by the MAD and SID. Our method significantly reduces the number of cases where debates drift from correct to incorrect answers, while increasing the number of beneficial corrections, i.e., debates that shift from wrong to correct outcomes, further demonstrating the high effectiveness of our method.

## 6 CONCLUSION

This work introduces SID, a multi-LLM debate framework that leverages self signals from the LLM generation process to improve both performance and efficiency. SID integrates two types of internal signals: model-level confidence, which enables early exit for confident agents, and token-level semantic focus, which compresses debate history by using attention scores to retain key points of disagreement. Experiments across diverse benchmarks with various LLMs and MLLMs demonstrate the high performance and efficiency of SID, underscoring the strong potential of leveraging internal model states as effective signals for guiding collaborative problem-solving. These findings point toward a promising direction for developing new paradigms in multi-agent systems.

## REPRODUCIBILITY STATEMENT

Significant efforts have been made to ensure the reproducibility of our results. The implementation details of our framework are described in the main manuscript (Section 4, Algorithm 1, and 5.1), including methods, baselines, benchmarks, model configurations, and evaluation settings. Additional implementation details and the full algorithm are provided in Appendix C. To facilitate faithful replication of our method, we include detailed descriptions of the key prompts and instruction formats in Table 4, Table 5, and Figure 4. We believe these materials are sufficient to enable reproducibility of our study.

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

## A    USE OF LARGE LANGUAGE MODELS (LLMS)

Large Language Models (LLMs) were used solely for language refinement and proofreading purposes. They were not involved in research ideation and methodology design. All scientific contributions and conceptual developments were carried out entirely by the authors. The LLM did not play a substantive role in shaping the research content and should not be considered a contributor.

## B    LIMITATION

Our method relies on internal model signals such as logits and attention maps, which limit direct applicability to public closed-source APIs. However, it remains well-suited for internal deployments of proprietary models, especially in multi-agent systems, and can serve as an intermediate reasoning layer prior to externalized API serving. Notably, many modern systems (e.g., GPT-5) already adopt multi-agent or tool-augmented architectures, making our approach broadly applicable and increasingly relevant.

## C    ALGORITHM AND IMPLEMENTATION

### C.1    SEMANTIC PRESERVATION

The semantic preservation module plays a key role in restoring the semantic cohesion from the top-$p$ selected sparse tokens based on a model's self-signals (*i.e.,* attention mechanism). Specifically, the method selects the most relevant textual spans based on attention distribution, but ensures that these selections are semantically coherent when mapped back to natural language. The algorithm below (Algorithm 2) shows the main pipeline, and the example (Figure 6) illustrates the comparison between without and with semantic preservation.

---

**Algorithm 2** Semantic-Preserving Compression

---

**Require:** Prompt text $x$ with marked spans FOCUS, DISCUSSION; offset map $\mathcal{O}$; Top-p selected attention score $\mathcal{C}$

**Ensure:** Compressed prompt $x'$
1: $\mathcal{U} \leftarrow$ EXTRACTUNITS($x$, DISCUSSION)                    ▷ Sentence/clause-level segments
2: $T \leftarrow$ TOKENIZER($x$)
3: $\mathcal{S} \leftarrow$ MAPTOKENSTOUNITS($\mathcal{C}, \mathcal{U}, \mathcal{O}$)
4: $x' \leftarrow$ REPLACESPAN($x$, DISCUSSION, $\mathcal{S}$)
5: **return** $x'$

---

We begin by extracting semantically coherent units (e.g., sentences or clauses) from the DISCUSSION span using lightweight parsing heuristics, including punctuation or newline segmentation. This yields a set of candidate text fragments $\mathcal{U}$.

We then use the Top-p selected attention score $\mathcal{C}$ (from Algorithm 1) to select the top-$p$ most relevant tokens from T. To preserve semantic interpretability, we map these selected tokens back to their enclosing segments in $\mathcal{U}$ using the token-to-text offset map $\mathcal{O}$. The resulting set of informative fragments $\mathcal{S}$ is used to replace the original DISCUSSION span, yielding a compressed prompt $x'$ that retains critical disagreement signals while discarding redundant or low-relevance content.

In the multi-modal setting (e.g., MLLMs), token offsets may shift due to image-text fusion. We mitigate this by anchoring to stable textual markers in the FOCUS span to adjust $\mathcal{O}$ and maintain alignment.

This compression module is integrated into the overall SID framework to support efficient and interpretable multi-agent reasoning under token or latency constraints

## C.2 PROMPT TEMPLATE

In multi-task evaluation settings, especially those involving factual or multiple-choice benchmarks, we observe that models frequently generate semantically correct answers but fail to conform to the expected output format. This discrepancy is particularly pronounced in open-ended LLMs, where prior supervised fine-tuning (SFT) phases may introduce implicit formatting preferences (e.g., `\boxed` in math domains).

To mitigate this, we prepend a task-specific *system prompt* that explicitly enforces the desired answer format. Our full prompting format is:

```
<system prompt> + <question content> + <output instruction>
```

This method proves especially helpful for models with weaker instruction-following capabilities (e.g., LLaMA3.1-8B) and significantly reduces post-hoc answer parsing failures. Another example is the GLM4.1-V thinking model. The default multiple choice response uses a special boxed token, such as `<|begin_of_box|>B<|end_of_box|>`. By emphasizing the answer returning with brackets in the system prompt, GLM4.1V thinking yields `<|begin_of_box|>(B)<|end_of_box|>`. This allows us to extract the result using brackets in a unified way. Table 4 lists the dataset-specific system prompts and the enforced answer formats used in our experiments.

Table 4: Dataset-specific system prompts and enforced output formats for answer extraction.

| Dataset | System Prompt (Instruction) | Expected Output Format (for answer parsing) |
|---|---|---|
| `MMLUpro` | You are a trivia expert who knows everything. You are tasked to answer the following question. Give your final answer in the format of `(X)`, e.g., `(A)`. | `(A)`, `(B)`, etc. |
| `Math` | You are a math expert. You are tasked to determine the answer to the following question. Give your final answer in the form of `\boxed{answer}` in the last sentence of your response, e.g., `\boxed{[1, 3]}`. | `\boxed{...}` |
| `GPQA` | You are an expert in graduate-level science and mathematics. You will be presented with challenging questions designed to test your reasoning abilities. Your last sentence should be "The correct answer is `(insert answer here)`." | "The correct answer is `(A)`." |
| `ScienceQA` | You are a trivia expert who knows everything. You are tasked to answer the following question. Give your final answer in the format of `(X)`, e.g., `(A)`. | `(A)`, `(B)`, etc. |
| `MMStar` | You are an expert in multimodal task understanding, and your task is to answer the following questions. Give your final answer in the format of (X), e.g., (A) | `(A)`, `(B)`, etc. |

Table 5: Dataset-specific output instruction prompts.

| Dataset | Output Instruction |
|---|---|
| `MMLUpro` | Give your final answer in the format of '(X)' |
| `Math` | Give your final answer in the form of `\\boxed{answer}` at the end of your response, e.g., `\\boxed{[1, 3]}`. |
| `GPQA` | Your last sentence should be 'The correct answer is (insert answer here).' e.g., The correct answer is (A). |
| `ScienceQA` | Give your final answer in the format of '(X)'. You should only give one answer. For example, the answer is (A). |
| `MMStar` | Give your final answer in the format of '(X)'. You should only give one answer. |

---

## Reasoning Augmentation Prompt

**<COT (zero-shot)>**
<Generation 1/1>
Let's think step by step.

**<IO>**
<Generation 1/1>
Please directly give your answer.

**<Self-Consistency>**
<Generation N/N, i.e., after N-rounds generation>
<majority vote among multiple responses>

**<Step-back prompting, SBP>**
<Generation 1/2>
You are an expert at structured reasoning. Your task is to extract the subject concepts and
principles involved in solving the problem. In this step you don't need to give you final answer,
just extract the concepts and principles.
<Get Phase 1 Response>

<Generation 2/2>
Learned concepts and principles:
{Phase 1 Response}
Solve the problem step by step with your reasoning path, according to the concepts and principles
you have learned.

**<Self-Contrast>**
<Generation 1/4>
Let's think step by step

<Generation 2/4>
Please generate an alternative solution to this problem using a different approach or reasoning
method.

<Generation 3/4>
Now compare your original solution with the alternative solution:
1. What are the key differences between the two approaches?
2. Which approach seems more reliable and why?
3. Can you identify any weaknesses in either approach?
4. Based on this comparison, what is your final answer?

<Generation 4/4>
Based on your comparison of the different approaches, provide your final answer.

Figure 4: Details of reasoning augmentation prompt.

In terms of question content, we strictly follow the previous work (Du et al., 2024; Liu et al., 2024c)
in parsing the question to the chat template.

Moreover, we list the reasoning augmentation prompt (Figure 4 used in our experiments. Notably,
Output Instructions should still be used after those prompts to enhance the ability to follow instruc-
tions.

## D    EXPANDED EXPERIMENTS

To further broaden our empirical coverage, we additionally evaluate SID on GPQA (Rein et al., 2023), a challenging benchmark targeting advanced science knowledge and reasoning. We follow the same evaluation protocol as in the main experiments and use GPT-OSS-20B as the base model, with SID-v as the default debate configuration.

Table 6: Results on the GPQA benchmark using GPT-OSS-20B under the same evaluation protocol as in the main text.

| Method (GPQA) | CoT | IO | MAD | SID |
|---|---|---|---|---|
| Accuracy (%) $\pm$ std | $43.2 \pm 1.7$ | $41.4 \pm 1.6$ | $52.9 \pm 2.0$ | $\mathbf{54.8 \pm 1.5}$ |

As shown in Table 6, SID achieves the best performance among all compared methods on GPQA, demonstrating that our SID framework generalizes to challenging scientific reasoning tasks beyond the benchmarks reported in the main paper.

Exchange-of-Thought (EoT) (Yin et al., 2023) derives a model-level confidence score post hoc from the final discrete answers (e.g., options A/B/C/D) across rounds and/or agents, for example, by measuring how frequently the most common answer appears. This effectively treats consistency of verbalized outputs as a proxy for confidence. To more concretely compare SID with EoT, we run both methods with LLaMA3.1-8B on the MMLUpro dataset. We follow the same evaluation protocol as in our main experiments and configure three agents and two debate rounds, computing the EoT confidence exactly as described in the original paper. The results are summarized in Table 7.

Table 7: Comparison with EoT on MMLUpro using LLaMA3.1-8B.

| Method | CoT | MAD | EoT | SID |
|---|---|---|---|---|
| Accuracy (%) | 39.5 | 41.5 | 42.7 | **46.8** |

As shown in Table 7, EoT improves over standard MAD, indicating that consistency-based aggregation can indeed be beneficial. SID, however, still yields a clear additional gain.

We further examine whether SID extends naturally to *heterogeneous* multi-model debate. To this end, we consider two base models on MMLUpro: LLaMA-3.1-8B (denoted as model A) and GPT-OSS-20B (denoted as model B). We instantiate homogeneous pairs (A+A and B+B) as well as a heterogeneous pair (A+B), and measure the accuracy of each agent's final-round answer under MAD and SID.

Table 8: Heterogeneous multi-model debate on MMLUpro with LLaMA-3.1-8B (A) and GPT-OSS-20B (B). For the heterogeneous A+B setting, we report the final accuracy of each agent (A/B).

| Setting | A+A MAD | A+A SID | B+B MAD | B+B SID | A+B MAD (A/B) | A+B SID (A/B) |
|---|---|---|---|---|---|---|
| Acc. (%) | 41 | 47 | 65 | 71 | 65 / 65 | 64 / **72** |

As shown in Table 8, SID consistently improves performance over MAD in the homogeneous settings: the accuracy of A+A increases from 41% to 47%, and B+B from 65% to 71%. This mirrors the trends observed in the main experiments and indicates that SID's model-level early exit and token-level semantic-focus mechanisms remain beneficial across different base models.

The heterogeneous configuration (A+B) provides additional insight. Under MAD, both A and B achieve roughly the same final accuracy (65%), suggesting that the weaker model A benefits from debating with the stronger model B, while the stronger model does not fully realize its potential within an unstructured debate protocol. Under SID, the stronger model B reaches 72% accuracy, which exceeds all homogeneous baselines (including B+B with MAD at 65% and even B+B with

SID at 71%), while model A remains competitive at 64%. These results indicate that SID can better exploit model complementarity in heterogeneous groups: the weaker model can still benefit from interaction, but the stronger model is less likely to be "dragged down" by unnecessary debate and can more reliably achieve (or slightly exceed) its best homogeneous performance.

Overall, this study suggests that SID is not only compatible with heterogeneous multi-model debate, but also capable of leveraging internal self-signals to coordinate agents of different strengths more effectively than standard MAD.

## E  MODEL-LEVEL CONFIDENCE ANALYSIS

In this section, we provide additional analyses of the model-level confidence signal used by SID, in order to clarify (i) how well it separates correct and wrong predictions, and (ii) what happens in over-confident failure cases.

Recall that for each agent, we compute a model-level confidence score from internal self-signals during generation. Concretely, given a sequence of output tokens $\{y_t\}_{t=1}^T$ and model log-probabilities $\log p_\theta(y_t \mid y_{<t}, x)$, we construct confidence metrics based on aggregated negative log-likelihood (NLL) and entropy over different token positions (e.g., all tokens, answer tokens, or reasoning tokens). These metrics are then calibrated via the vocabulary-adaptive threshold described in Section 4.1 to decide whether an agent should exit the debate early.

**Separation between correct and wrong predictions.**    Figures 7–16 in the appendix, for multiple datasets and models, the empirical distributions of our confidence metrics for *correct* (C) vs. *wrong* (W) answers, together with significance tests. Across MMLU-Pro, MATH subsets, ScienceQA, MMStar, and GPQA, we consistently observe that the C group exhibits noticeably *lower* NLL / entropy than the W group, and these differences are statistically significant in most settings. In parallel, the correction-flow plots in the same figures show that, under SID, the number of correct-to-wrong (C→W) transitions across rounds is reduced, while wrong-to-correct (W→C) transitions are maintained or increased compared to MAD. Taken together, these results indicate that our confidence estimate, although not perfectly calibrated, is a useful ranking signal for debate scheduling: high-confidence states are more likely to be correct and more likely to remain correct under SID.

**Over-confident errors and failure cases.**    We also explicitly examine cases where the model-level confidence is high but the final answer may be incorrect. Figures 17–20 provide qualitative examples of such failure cases under our early-exit policy. In most of these examples, the intermediate reasoning trajectory is largely sensible (e.g., correctly recalling definitions or setting up equations), but the model makes a local slip in the last step, such as an arithmetic mistake or an incorrect option mapping. Because our confidence is derived from token-level log-probabilities aggregated over the entire reasoning sequence, these mostly plausible trajectories can still yield low NLL / entropy even when the final box answer is wrong. These examples illustrate the limitations of our signal: it is not an oracle and over-confident errors do occur.

However, our quantitative analyses show that such over-confident wrong cases are relatively rare compared to the large mass of high-confidence correct predictions. In addition, the correction-flow statistics indicate that SID reduces harmful C→W transitions overall, while preserving beneficial W→C transitions. Thus, even though the confidence signal can occasionally fail, on balance it enables the early-exit mechanism to (i) protect many high-confidence correct answers from being overturned by noisy debate and (ii) avoid spending additional tokens on debates that are unlikely to change the outcome.

In summary, our model-level confidence should be viewed as a ranking heuristic derived from internal self-signals. Empirically, it exhibits a clear and statistically meaningful separation between correct and wrong predictions across benchmarks, and it leads to fewer C→W transitions and more efficient use of debate rounds when integrated into SID. This supports its use as a practical gating signal for deciding when to continue or terminate multi-agent debate.

# F  MORE RESULTS

In this section, we list many visualization results to illustrate the effectiveness of our SID methods.

Figure 6 Example comparison between w/o semantic preservation (red, brute-force token selection) and w/ semantic preservation (green, semantically coherent expansion). It can be observed that token-level semantic focus deletes irrelevant points (for example, point 2 in the solution was deleted), and our semantic preservation retains the semantic cohesion from the selected tokens.

A series of model-level confidence examples below can demonstrate the stable early exit threshold in the same model, and the statistical significance between the correct and wrong groups. Moreover, we also provide the correction flow from the first round to the last round.

Figure 7 Top: Model-level Confidence result on the Math Algebra dataset with LLaMA3.1-8B. Bottom: Correction flow with 8 deltas of confidence metrics. Unc: the answer remains unchanged between the first round and the last round, C2W: correct to wrong, W2C: wrong to correct.

Figure 8 Model-level Confidence result on the Math Counting and Probability dataset with LLaMA3.1-8B.

Figure 9 Model-level Confidence result on the Math Geometry dataset with LLaMA3.1-8B.

Figure 10 Model-level Confidence result on the Math Intermediate Algebra dataset with LLaMA3.1-8B.

Figure 11 Model-level Confidence result on the Math Number Theory dataset with LLaMA3.1-8B.

Figure 12 Model-level Confidence result on the Math Prealgebra dataset with LLaMA3.1-8B.

Figure 13 Model-level Confidence result on the Math Precalculus dataset with LLaMA3.1-8B.

Figure 14 Model-level Confidence result on the MMStar dataset with GLM4.1V.

Figure 15 Model-level Confidence result on the MMLUpro dataset with GPT-OSS-20B. Bottom: Correction flow with 8 deltas of confidence metrics.

In addition, a number of model-level early exit cases are provided here to show the confident and overconfident cases. It can be observed that the model partially analyzes the problem in overconfident cases.

Figure 17 Examples of model-level early exit cases in the MMLUpro dataset.

Figure 18 Examples of model-level early exit cases in the ScienceQA dataset.

Figure 19 Examples of model-level early exit cases in the Math dataset.

Figure 20 Examples of model-level early exit cases in the GPQA dataset.

Subsequently, Figure 21 and Figure 22 display that the token-level semantic focus module compresses the contents and assists agents in correcting their answers.

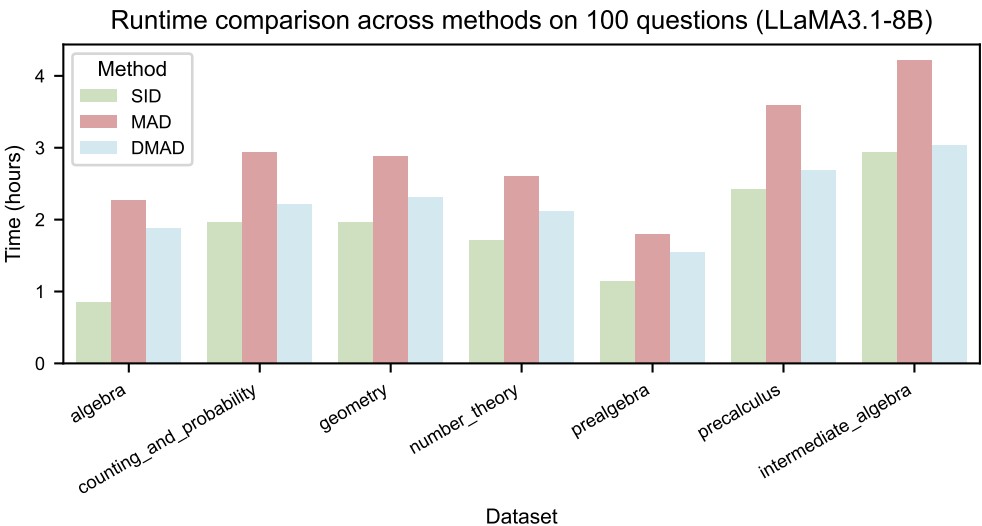

Figure 5: Running time comparison on math datasets with LLaMA3.1-8B, running on single A100 80GB GPU

## Examples of Semantic Preservation

**<Selected Text: w/o Semantic Preservation>**

One agent solution:

```To solve this, let's break it down by step:

. Both parents have the same meaning they are both they can only that also zy they can only).

**3.** Since are but not).

**4** the probability that each of children will bezy is1or %), because they will only from both parents.

So, the correct answer is (F) **1**.

```

One agent solution:

```To solve this let's consider the possibleotypes andotypes of the and both parents are we can their the a

**2** Theotypes of )

**3** parents the possible can

**4** The probability of each child is the of which is because each has of probability of2).

So, the probability that each children will be affected equals **4** or **25**%. Therefore, the answer is (I).

```

**<Selected Text: w/ Semantic Preservation>**

One agent solution:

```To solve this, let's break it down step by step:

**1.** Both parents have the same autosomal recessive disorder, meaning they are both homozygous recessive (rr) for the disorder. When two homozygous recessive parents (rr) have children, they can only produce offspring that are also homozygous recessive (rr) because they can only contribute their recessive alleles (r).

**3.** Since the disorder is autosomal recessive, the parents are carriers of the recessive allele (r) but do not express the disorder themselves because they are homozygous recessive (rr).

**4.** Therefore, the probability that each of their children will be affected (homozygous recessive, rr) is **1** (or **100**%), because they will only inherit the recessive allele from both parents. So, the correct answer is (F) **1**.```

One agent solution: ```To solve this, let's consider the possible genotypes and phenotypes of the parents and their offspring. Since both parents are affected by an autosomal recessive disorder, we can represent their genotype as `'rr'` (assuming the disorder is caused by a recessive allele `'r'` and the dominant allele is `'R'`).

**2.** The possible genotypes of their offspring are: - Rr (carrier) - Rr (carrier) **3.** Since both parents are affected, the possible genotypes of the parents can be rr.

**4.** The probability of each child being affected (rr) is the probability of inheriting two recessive alleles, which is **1/4**, because each parent has a **1/2** chance of contributing a recessive allele, and the probability of both contributing a recessive allele is (**1/2**) * (**1/2**) = **1/4**. So, the probability that each of their children will be affected equals **1/4** or **25**%. Therefore, the answer is (I).```

**Legend: The red and green shaded text is an example pair for the w/o Semantic Preservation and w/ Semantic Preservation comparison.**

Figure 6: Example comparison between w/o semantic preservation (red, brute-force token selection) and w/ semantic preservation (green, semantically coherent expansion)

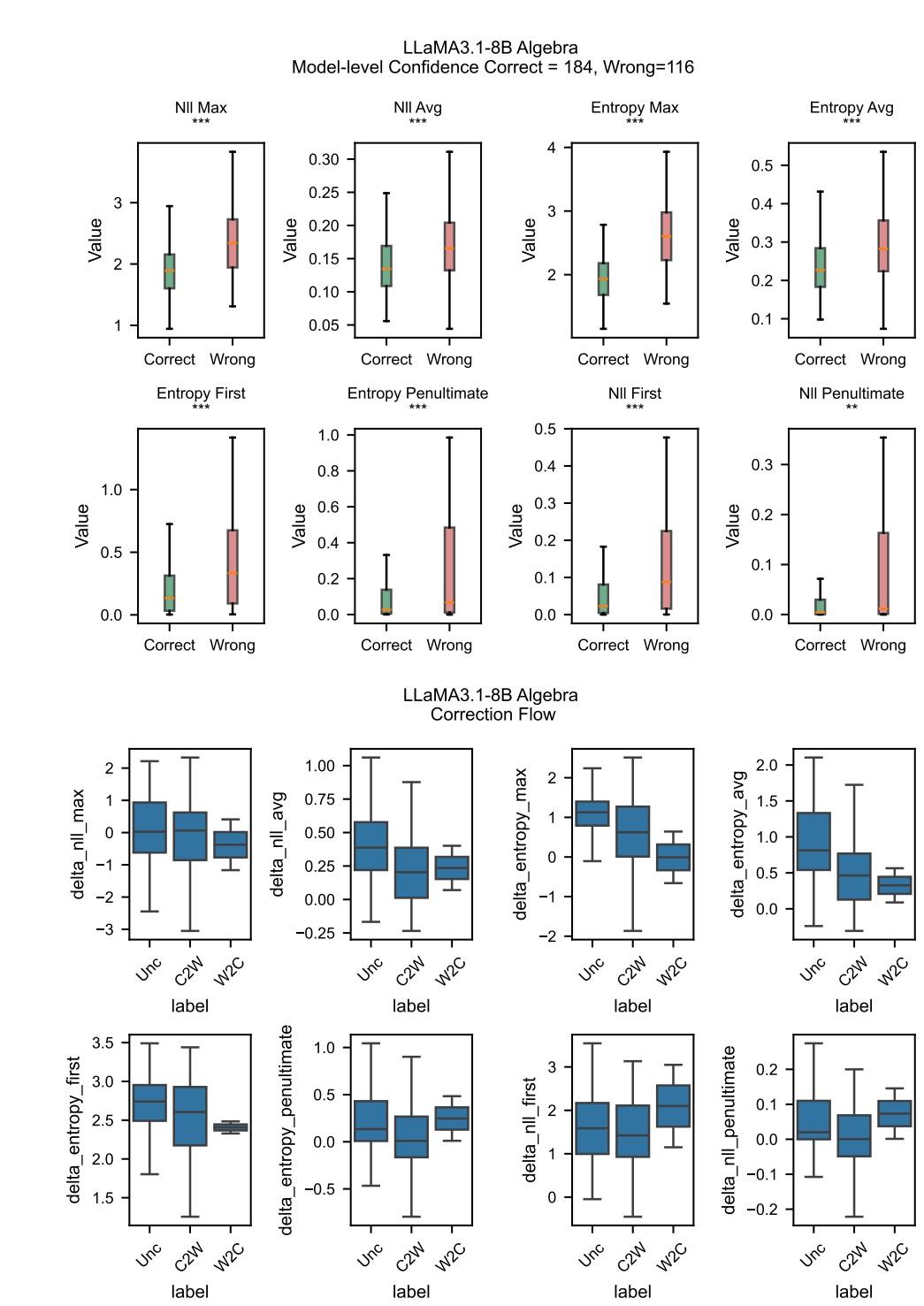

Figure 7: Top: Model-level Confidence result on the Math Algebra dataset with LLaMA3.1-8B. Bottom: Correction flow with 8 deltas of confidence metrics. Unc: the answer remains unchanged between the first round and the last round, C2W: correct to wrong, W2C: wrong to correct.

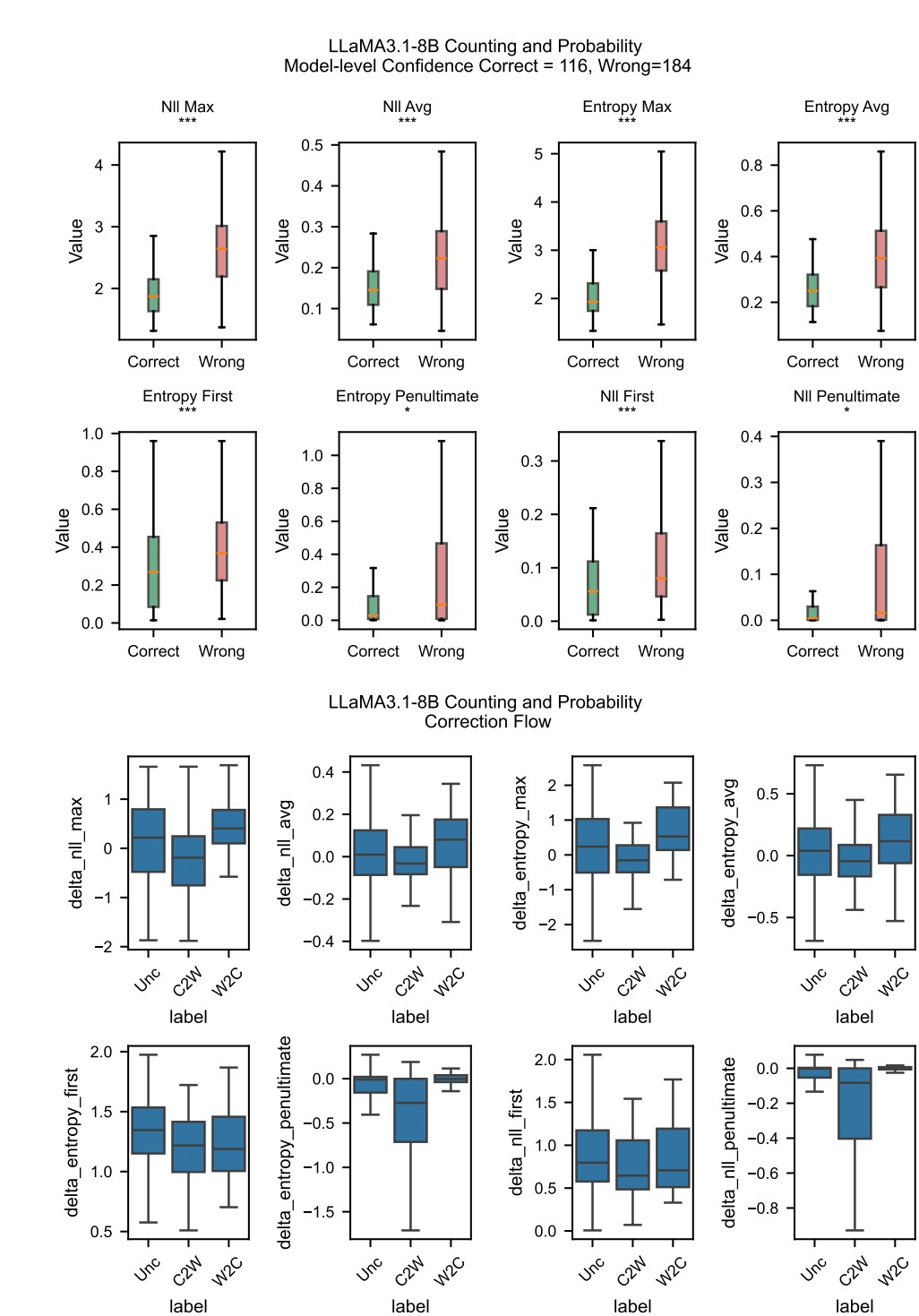

Figure 8: Top: Model-level Confidence result on the Math Counting and Probability dataset with LLaMA3.1-8B. Bottom: Correction flow with 8 deltas of confidence metrics. Unc: the answer remains unchanged between the first round and the last round, C2W: correct to wrong, W2C: wrong to correct.

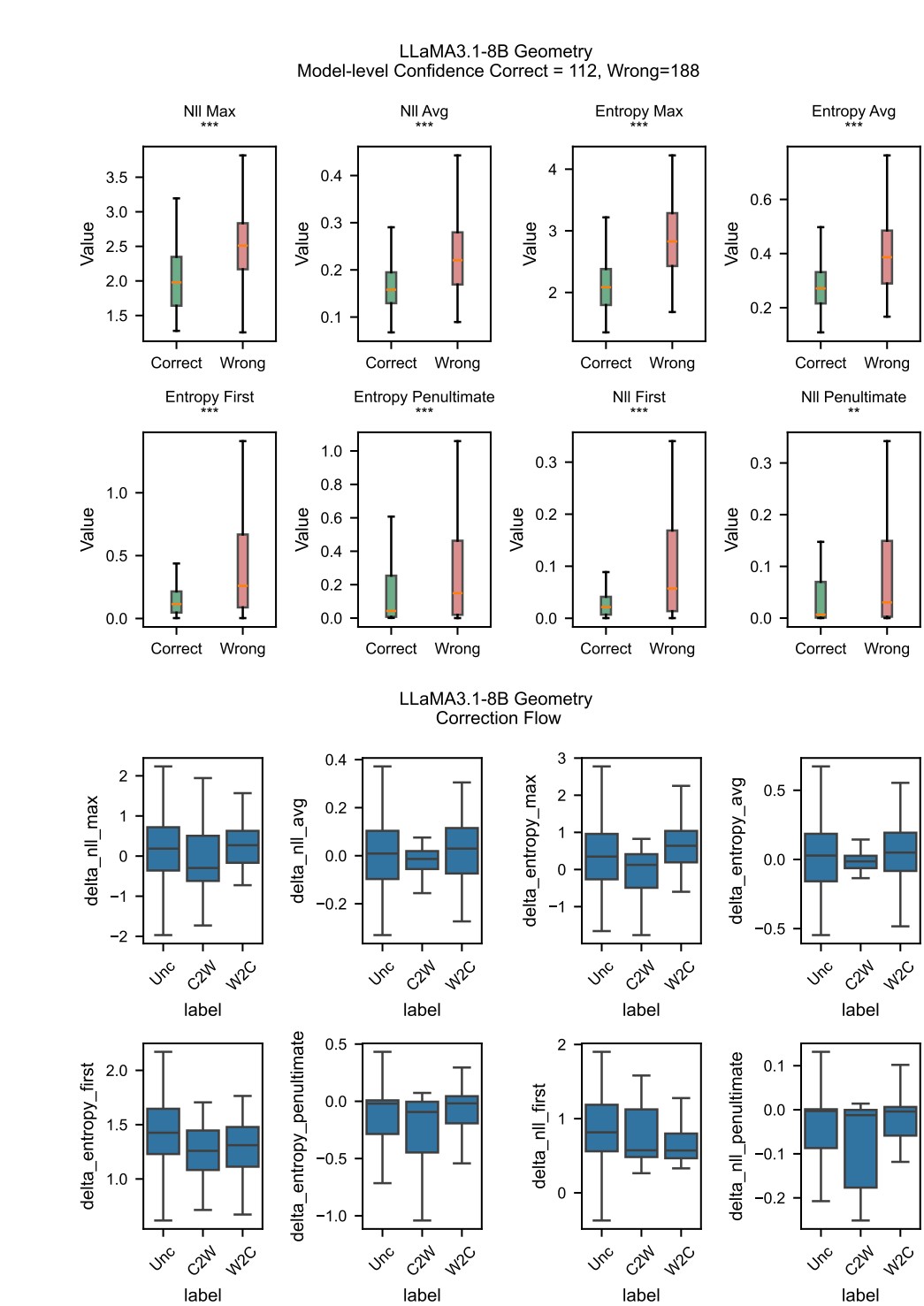

Figure 9: Top: Model-level Confidence result on the Math Geometry dataset with LLaMA3.1-8B. Bottom: Correction flow with 8 deltas of confidence metrics. Unc: the answer remains unchanged between the first round and the last round, C2W: correct to wrong, W2C: wrong to correct.

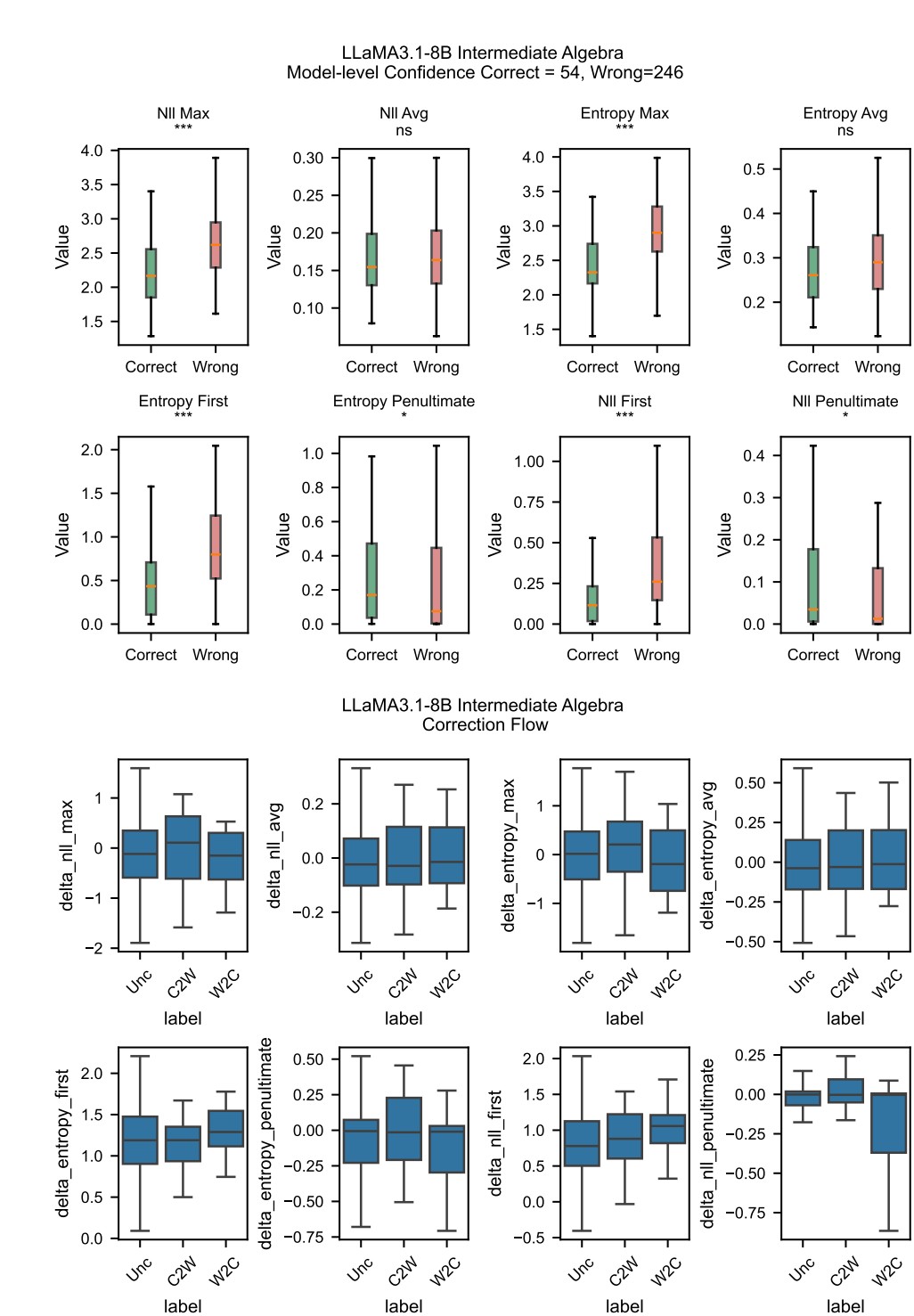

Figure 10: Top: Model-level Confidence result on the Math Intermediate Algebra dataset with LLaMA3.1-8B. Bottom: Correction flow with 8 deltas of confidence metrics. Unc: the answer remains unchanged between the first round and the last round, C2W: correct to wrong, W2C: wrong to correct.

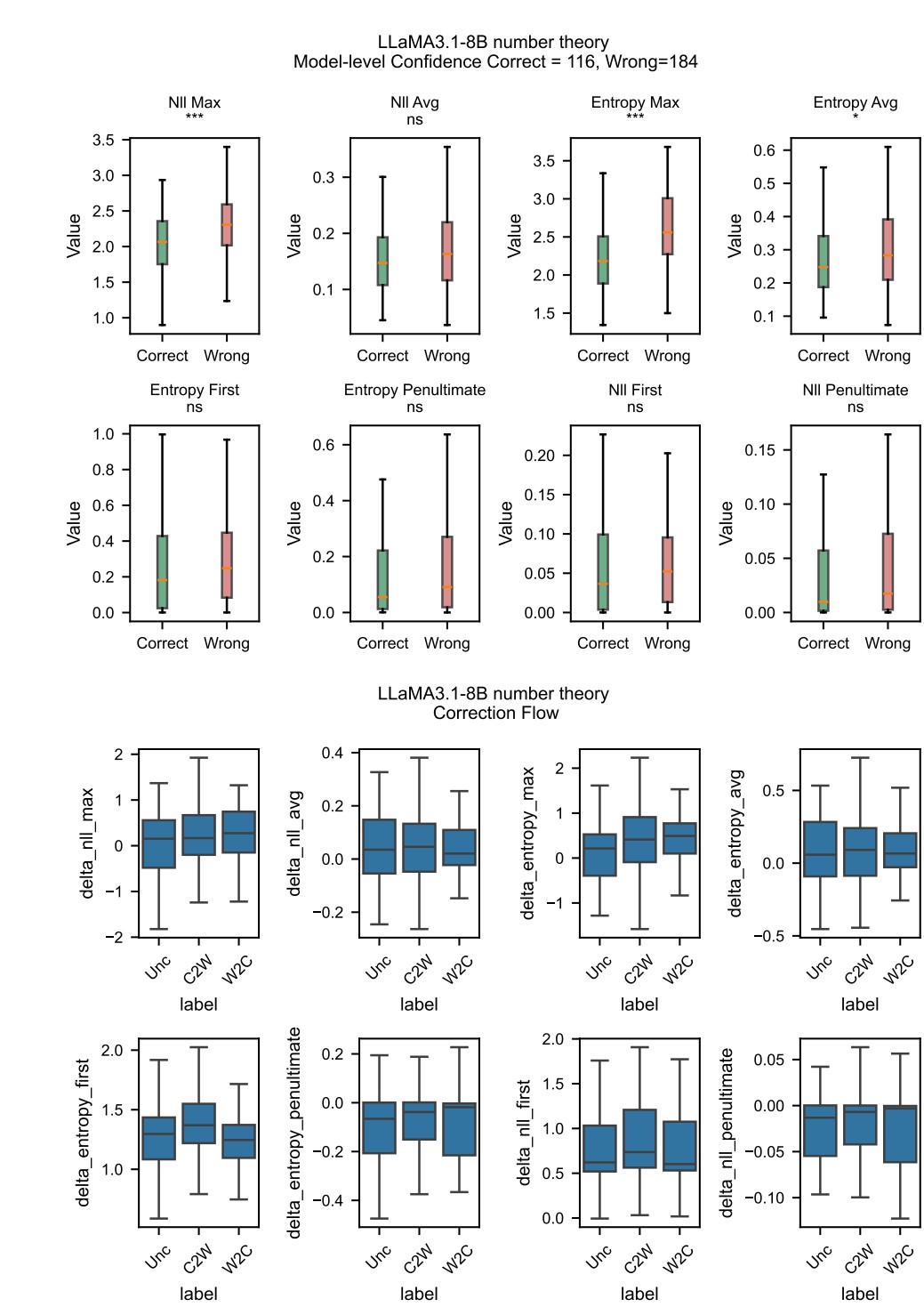

Figure 11: Top: Model-level Confidence result on the Math Number Theory dataset with LLaMA3.1-8B. Bottom: Correction flow with 8 deltas of confidence metrics. Unc: the answer remains unchanged between the first round and the last round, C2W: correct to wrong, W2C: wrong to correct.

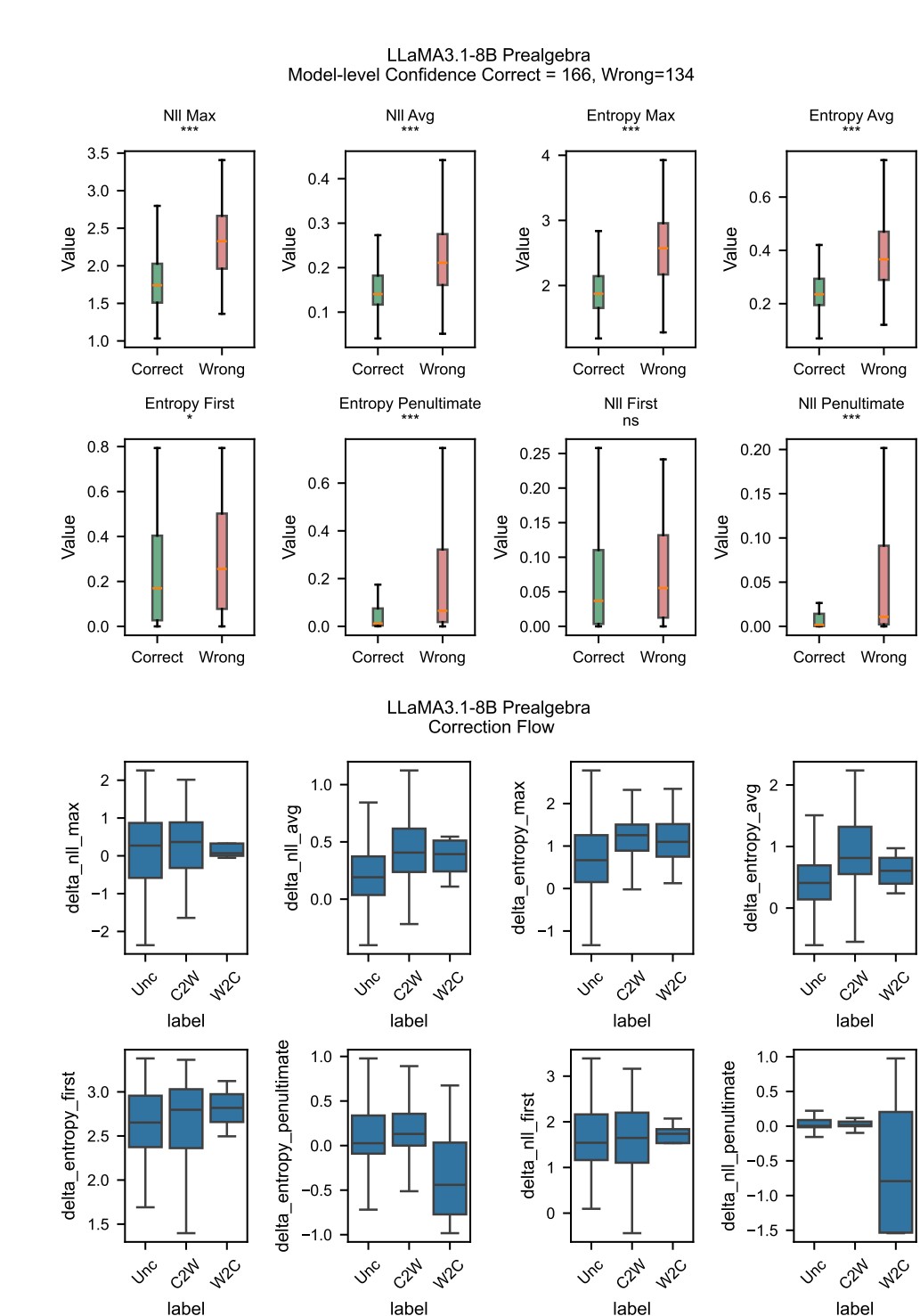

Figure 12: Top: Model-level Confidence result on the Math Prealgebra dataset with LLaMA3.1-8B. Bottom: Correction flow with 8 deltas of confidence metrics. Unc: the answer remains unchanged between the first round and the last round, C2W: correct to wrong, W2C: wrong to correct.

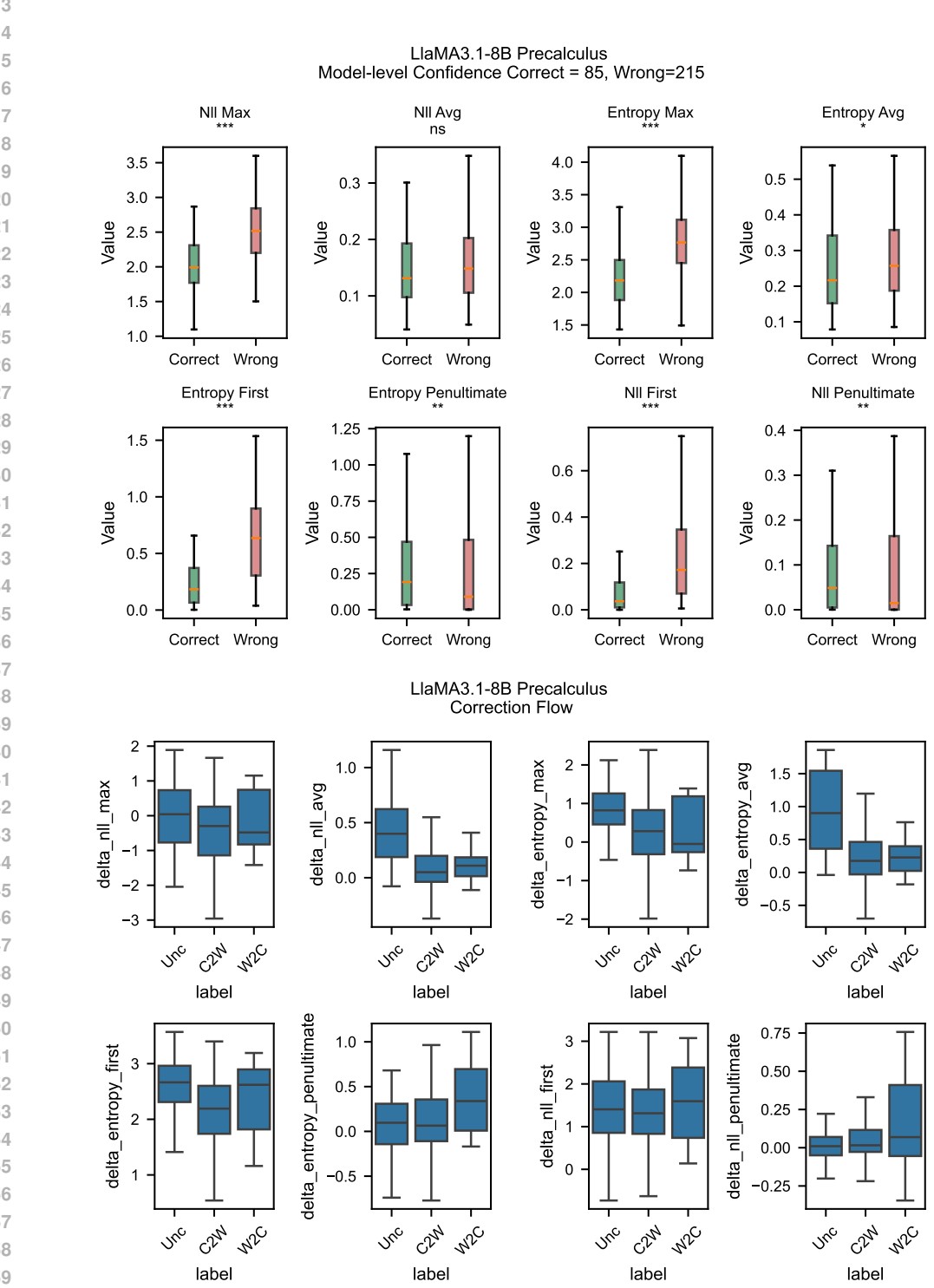

Figure 13: Top: Model-level Confidence result on the Math Precalculus dataset with LLaMA3.1-8B. Bottom: Correction flow with 8 deltas of confidence metrics. Unc: the answer remains unchanged between the first round and the last round, C2W: correct to wrong, W2C: wrong to correct.

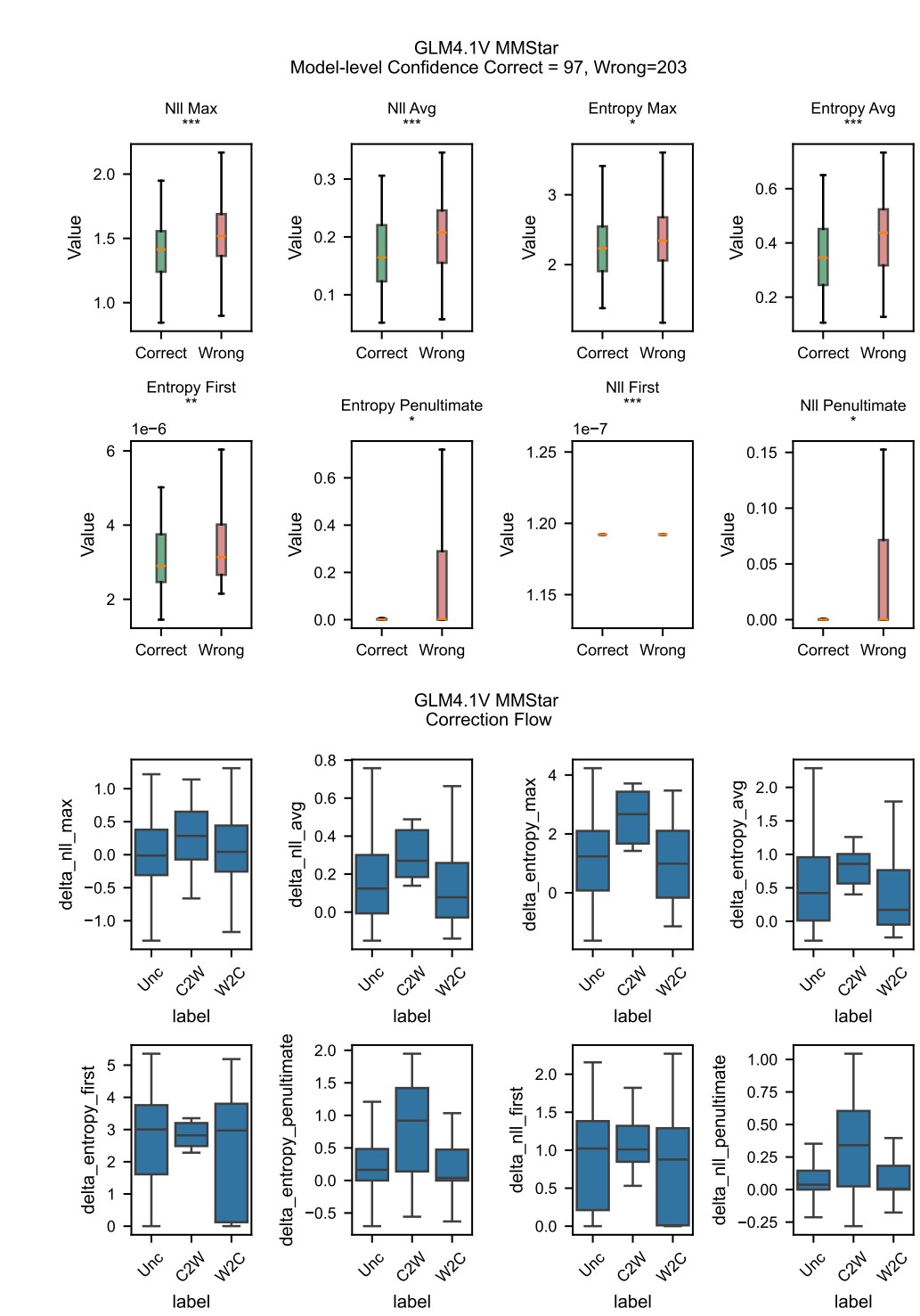

Figure 14: Top: Model-level Confidence result on the MMStar dataset with GLM4.1V. Bottom: Correction flow with 8 deltas of confidence metrics. Unc: the answer remains unchanged between the first round and the last round, C2W: correct to wrong, W2C: wrong to correct.

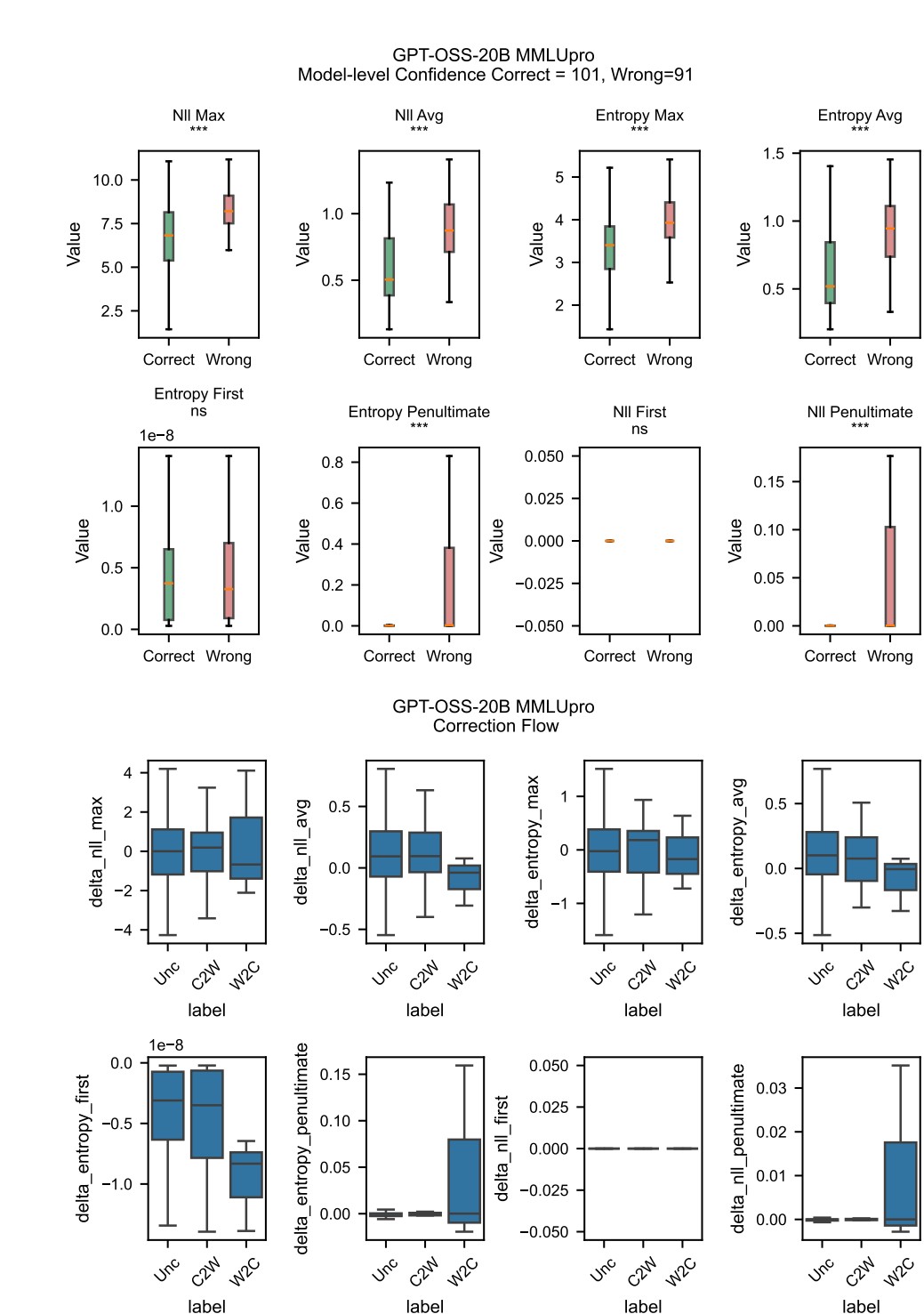

Figure 15: Top: Model-level Confidence result on the MMLUpro dataset with GPT-OSS-20B. Bottom: Correction flow with 8 deltas of confidence metrics. Unc: the answer remains unchanged between the first round and the last round, C2W: correct to wrong, W2C: wrong to correct.

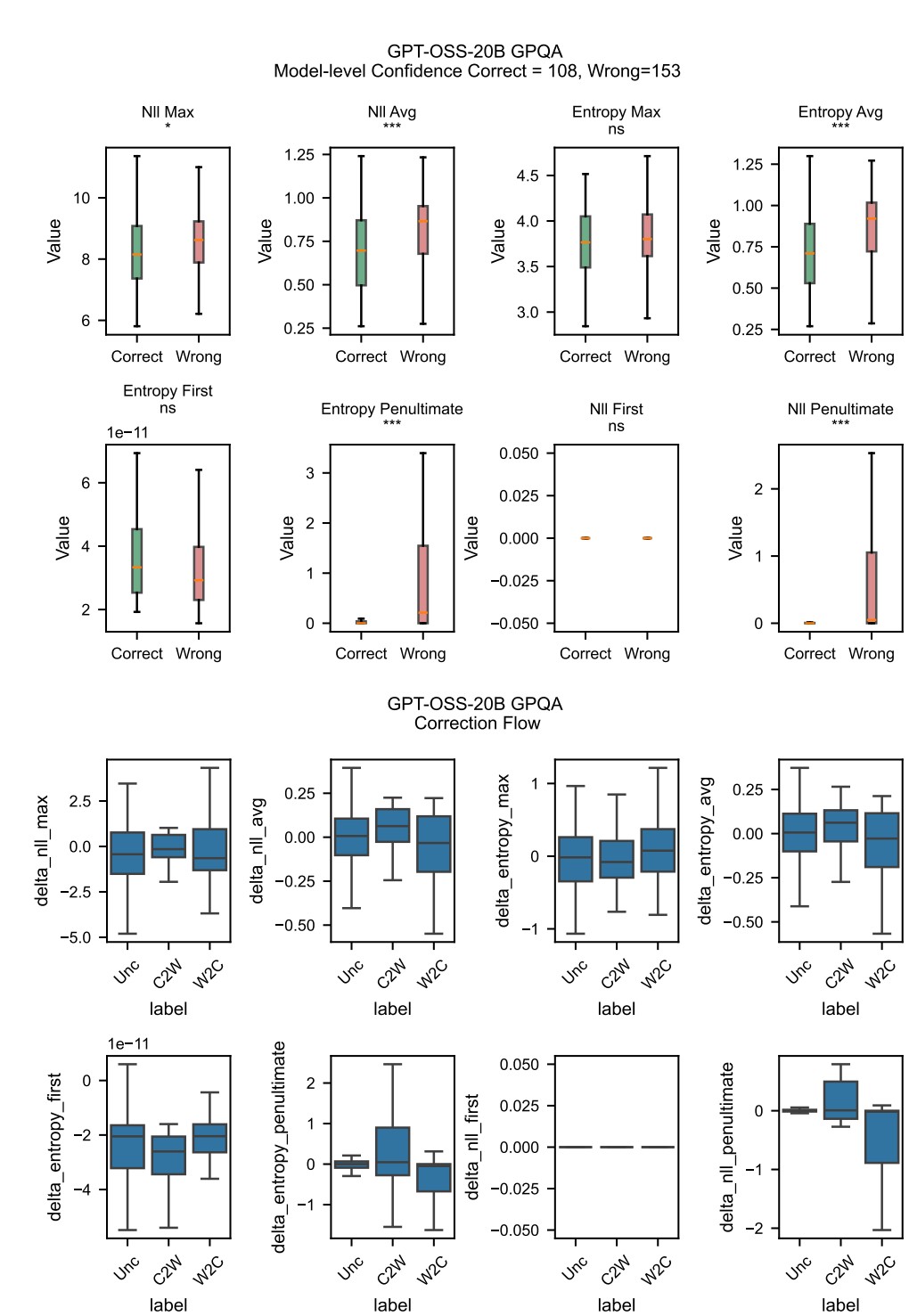

Figure 16: Top: Model-level Confidence result on the GPQA dataset with GPT-OSS-20B. Bottom: Correction flow with 8 deltas of confidence metrics. Unc: the answer remains unchanged between the first round and the last round, C2W: correct to wrong, W2C: wrong to correct.

## Examples of Model-Level Confidence Exit on MMLUpro

**<Question, Correct Answer: B>**
Can you answer the following question? If a freight train travels at a speed of 20 miles per hour for 6 hours, how far will it travel?: (A) 90 miles, (B) 120 miles, (C) 150 miles, (D) 140 miles, (E) 26 miles, (F) 100 miles, (G) 80 miles, (H) 110 miles, (I) 160 miles, (J) 12 miles.

**<Agent Answer: B> ✅**
To find the distance traveled by the freight train, we need to multiply its speed by the time it traveled.\n\nSpeed = 20 miles per hour\nTime = 6 hours\n\nDistance = Speed x Time\n= 20 miles/hour x 6 hours\n= 120 miles\n\nSo, the freight train will travel 120 miles. \n\nThe correct answer is (B).

**<Question, Correct Answer: E>**
Can you answer the following question? A driver and passenger were driving to work in the former's automobile one morning. As they were traveling at a speed of 20 m. p. h. (which was within the posted speed limit), the passenger suddenly pointed to an overturned vehicle along the side of the highway and said, "Look at that car upside down. " The driver turned to look at the overturned vehicle. As he was looking toward the side of the road, the driver failed to see an abandoned vehicle with a flat tire in the highway about 200 feet in front of his approaching auto. Seconds later, the driver crashed into the rear of the abandoned auto and was injured. The jurisdiction has a relevant comparative negligence statute in effect. If the driver asserts a claim against the owner of the abandoned auto, the most likely result is that the driver will: (A) recover nothing, because the passenger's act was a supervening cause., (B) recover only a portion of damages, because both the driver and the owner of the abandoned auto were negligent ., (C) recover nothing, because he was distracted while driving., (D) recover all of his damages, because the owner of the abandoned auto was negligent., (E) recover only a portion of damages, because the abandoned auto was in plain view., (F) recover all of his damages, because the abandoned auto was a traffic hazard., (G) recover all of his damages, because the defendant created a dangerous condition., (H) recover nothing, because he was driving at a speed that didn't allow him to stop in time., (I) recover nothing, because he had the last clear chance to avoid the collision..

**<Agent Answer: B> ❌ [But Judged by Gemini and GPT, both B and E are correct.]**
(B) To determine the most likely result, let's break down the situation step by step:\n\n1. The driver was driving at a speed of 20 m.p.h. within the posted speed limit.\n2. The passenger pointed out the overturned vehicle, distracting the driver and causing him to take his eyes off the road.\n3. As a result, the driver failed to see the abandoned vehicle with a flat tire in front of him.\n4. The driver crashed into the abandoned vehicle.\n\nGiven the comparative negligence statute, the jurisdiction likely follows a modified comparative negligence system. In this system, a plaintiff can recover damages even if they were partially at fault, but their recovery is reduced by the percentage of their own negligence.\n\nThe driver's distraction caused by the passenger's comment likely contributed to the accident, making him partially responsible for the collision. However, the owner of the abandoned auto also created a hazard by leaving the vehicle on the road. Therefore, both parties contributed to the accident.\n\nThe most likely result is that the driver will recover only a portion of damages, as both the driver and the owner of the abandoned auto were negligent.

Figure 17: Examples of model-level early exit cases on MMLUpro datasets. The exit questions are correctly answered in principle.

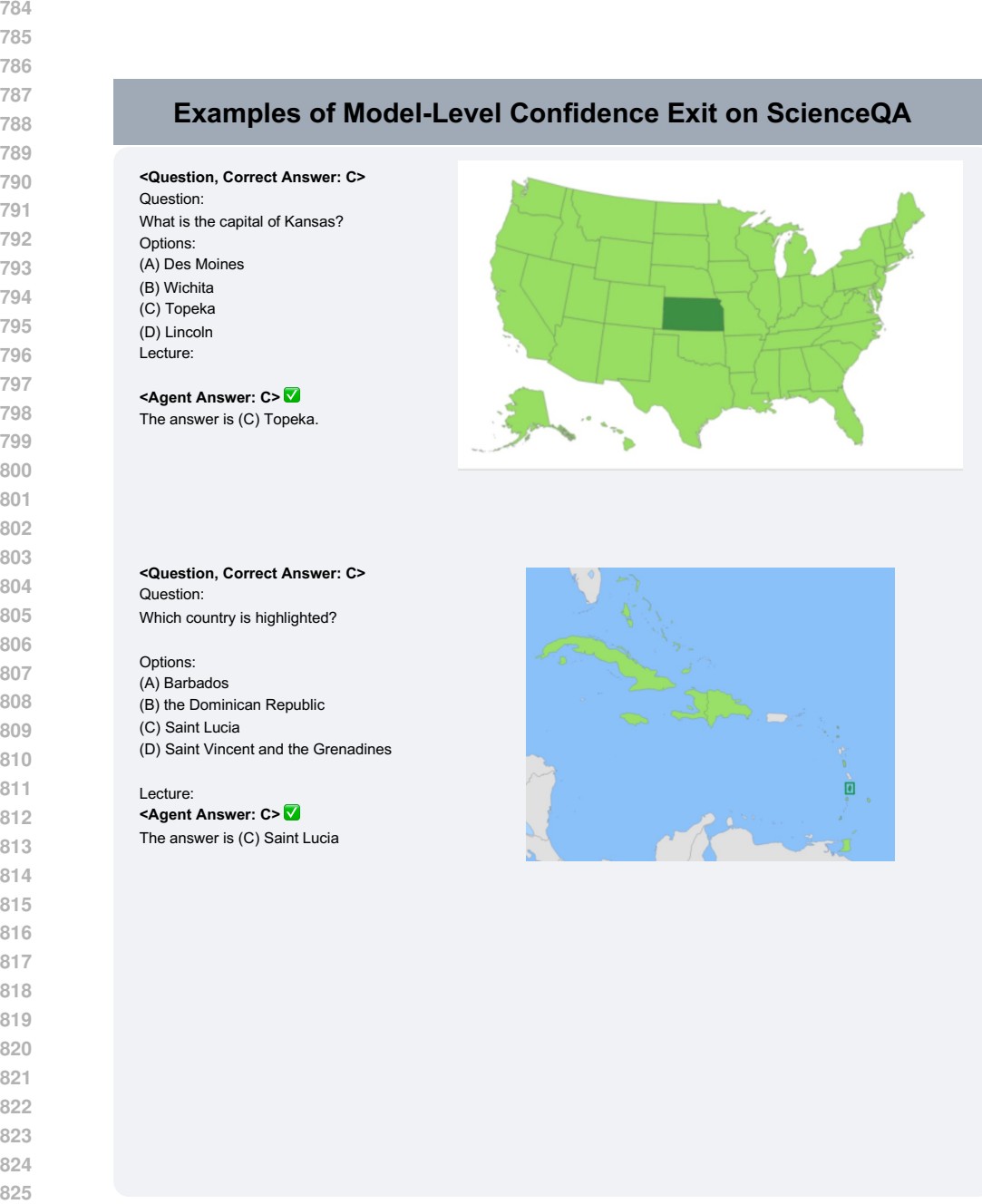

Figure 18: Examples of model-level early exit cases on ScienceQA datasets. The exit questions are correctly answered in principle.

## Examples of Model-Level Confidence Exit on Math

**<Question, Correct Answer: [1, 4]>**

Question:

Given a mathematics problem, determine the answer.

Question:Compute the domain of the real-valued function \[f(x)=\sqrt{1-\sqrt{2-\sqrt{x}}}.\]

Your final answer should be in the form \boxed{answer}, at the end of your response.

**<Agent Answer: [1, 4]>** ✅

To find the domain of the function $f(x)=\\sqrt{1-\\sqrt{2-\\sqrt{x}}}$, we need to ensure that the expression under the square roots is non-negative.\n\nFirst, we start with the innermost expression, $\\sqrt{x}$. Since the square root of a number is non-negative, we know that $x \\geq 0$.\n\nNext, we consider the expression $2-\\sqrt{x}$. For this to be non-negative, we need $2-\\sqrt{x} \\geq 0$. Solving for $\\sqrt{x}$, we get $\\sqrt{x} \\leq 2$, which implies $x \\leq 4$.\n\nFinally, we consider the expression $1-\\sqrt{2-\\sqrt{x}}$. For this to be non-negative, we need $1-\\sqrt{2-\\sqrt{x}} \\geq 0$. Solving for $\\sqrt{2-\\sqrt{x}}$, we get $\\sqrt{2-\\sqrt{x}} \\leq 1$, which implies $2-\\sqrt{x} \\leq 1$. Simplifying, we get $\\sqrt{x} \\geq 1$, which implies $x \\geq 1$.\n\nCombining the conditions, we have $1 \\leq x \\leq 4$. Therefore, the domain of the function is $\\boxed{[1, 4]}$.'

**<Question, Correct Answer: 7>**

Given a mathematics problem, determine the answer.

Question:Evaluate $\left\lceil{\frac32}\right\rceil^2+\left\lceil{\left(\frac32\right)^2}\right\rceil$.

Your final answer should be in the form \boxed{answer}, at the end of your response.

**<Agent Answer: 9>** ❌

To evaluate this expression, we need to follow the order of operations (PEMDAS):\n\n1. Evaluate the expressions inside the parentheses: $\\left(\\frac32\\right)^2 = \\frac94 = 2.25$\n2. Evaluate the ceiling function of the result: $\\left\\lceil 2.25 \\right\\rceil = 3$\n3. Evaluate the expression inside the other set of parentheses: $\\frac32$\n4. Evaluate the ceiling function of the result: $\\left\\lceil \\frac32 \\right\\rceil = 2$\n5. Square the result of step 3: $\\left(\\frac32\\right)^2 = \\frac94 = 2.25$\n6. Add the results of steps 2 and 5: $3 + 2.25 = 5.25$\n7. Evaluate the ceiling function of the result of step 6: $\\left\\lceil 5.25 \\right\\rceil = 6$\n\nNow, add the results of steps 2 and 7: $3 + 6 = 9$\n\nTherefore, the final answer is \\boxed{9}.'

Figure 19: Examples of model-level early exit cases on Math datasets. The bottom overconfident case shows the model partially analyzes the question, potentially leading to a high confidence.

## Examples of Model-Level Confidence Exit on GPQA

**<Question, Correct Answer: B>**

Can you answer the following question? In the vacuum, we have the following mixture
$\left|\nu_{i}\left(x\right)\right\rangle =e^{ip_{1}x}\cos\theta\left|\nu_{1}\right\rangle +e^{ip_{2}x}\sin\theta\left|\nu_{2}\right\rangle $
where $i=e,\mu,\nu, \theta$ the mixing angle, and $\nu_{1}$ and $\nu_{2}$ are the basis of mass eigenstates.
At what value of the mixing angle we will obtain the transition probability $P\left(\nu_{e}\rightarrow\nu_{\mu}\right)=1$.
Choices: (A) pi/2 (B) pi/4 (C) pi (D) pi/3.

**<Agent Answer: B> ✅**

analysis:We need to analyze neutrino flavor mixing: initial state is electron neutrino? They give general state |v_i(x)> = e^{i p1 x} cos θ |v1> + e^{i p2 x} sin θ |v2>. The probability of transition from ve to vμ = sin^2 2θ sin^2(Δm^2 L / (4E)). For maximum probability 1, we need sin^2 2θ = 1 AND sin^2 phase term = 1 (i.e., argument = π/2 + nπ). But question asks at what mixing angle we will obtain the probability 1 independent of oscillation? If you want probability 1, you need perfect mixing: θ=π/4 gives maximal mixing so sin^2 2θ = sin^2(π/2)=1. So answer (B) π/4. So "The correct answer is (B)".assistantfinalIn a two-flavor framework the transition probability is \n\n\[\nP(\\nu_e\\rightarrow \\nu_\\mu)=\\sin^2 2\\theta \\,\\sin^2\\!\\left(\\frac{\\Delta m^2 L}{4E}\\right),\n\\]\n\nwhere \\(\\theta\\) is the mixing angle and the second sine factor contains the usual oscillation phase. \nTo make the probability equal to unity for some distance \\(L\\) we need both factors to be unity. \nThe phase factor can become unity for suitable \\(L\\) (e.g. when \\(\\frac{\\Delta m^2 L}{4E} = \\pi/2\\)). \nThe remaining requirement is therefore \n\n\[\n\\sin^2 2\\theta = 1 \\;\\;\\;\\Longrightarrow\\;\\; \\sin 2\\theta = \\pm 1 \\;\\;\\;\\Longrightarrow\\;\\; 2\\theta = \\frac{\\pi}{2}\\;( \\text{mod }\\pi).\n\]\n\nThus \n\n\[\n\\theta = \\frac{\\pi}{4} \\quad (\\text{mod }\\tfrac{\\pi}{2}).\n\]\n\nAmong the provided options, the value that gives complete mixing is \\(\\theta = \\pi/4\\). \nTherefore the correct choice is (B).\n\nThe correct answer is (B).'

**<Question, Correct Answer: B>**

Can you answer the following question? In a specific region of the sky, astronomers have observed that the number of stars varies with parallax as 1/plx^5. How does the number of stars in that region of the sky change with distance (per unit range of distance, r)? Choices: (A) ~ r^4 (B) ~ r^3 (C) ~ r^2 (D) ~ r^5.

**<Agent Answer: B> ✅**

analysisWe need to understand relationship between parallax and distance: parallax p (in arcseconds) approx 1/d(pc) maybe? For small parallax: p = 1/d in arcsec if distance in parsecs. So d = 1/p. Number of stars varying as 1/p^5. Number of stars per parallax interval? They say number of stars varies with parallax as 1/plx^5. That implies count N(p) ~ 1/p^5. We want how number per unit range of distance r behaves: N(r) per unit distance? The distribution of stars per unit distance range: dN/dr. The relation between p and r: p = 1/r (assuming p in arcsec, r in parsecs). So r = 1/p. So p = 1/r. So N(p) ~ 1/p^5. But N(p) is number in certain parallax interval maybe? We need convert to N(r). Suppose number per parallax interval dp is proportional to 1/p^5. The number per unit r: dN/dr = dN/dp * dp/dr. dN/dp = k * 1/p^5. dp/dr = d(1/r)/dr = -1/r^2. We take magnitude: |dp/dr| = 1/r^2. Express p in terms of r: p = 1/r. So (1/p^5) = 1/( (1/r)^5 ) = r^5. So dN/dp = k * r^5. Multiply by |dp/dr|=1/r^2 gives dN/dr = k * r^5 * (1/r^2) = k * r^3. So number per unit distance ~ r^3. So answer (B). That matches typical uniform space density: N(r) ~ r^3. So The correct answer is (B).assistantfinalThe relationship between parallax \\(p\\) and distance \\(r\\) is \\(p = 1/r\\) (with \\(p\\) in arcseconds and \\(r\\) in parsecs).\n\nThe number of stars as a function of parallax is given as \n\[\n\\frac{dN}{dp}\\propto \\frac{1}{p^5}.\n\]\nTo find the differential number per unit distance we use the chain rule\n\[\n\\frac{dN}{dr}= \\frac{dN}{dp}\\,\\frac{dp}{dr}.\n\]\n\nFrom \\(p = 1/r\\), we have \\(dp/dr = -1/r^2\\); taking the magnitude, \\( |dp/dr| = 1/r^2\\). \nSubstitute \\(p=1/r\\) into \\(dN/dp\\):\n\[\n\\frac{dN}{dp}\\propto \\frac{1}{(1/r)^5} = r^5.\n\]\n\nThus,\n\[\n\\frac{dN}{dr}\\propto r^5 \\times \\frac{1}{r^2} = r^3.\n\]\n\nHence, the number of stars per unit range of distance \\(r\\) scales as \\(r^3\\).\n\nThe correct answer is (B).'} **(We did not find negative cases when using model-level early exit.)**

Figure 20: Examples of model-level early exit cases on GPQA datasets, where we did not find negative cases (*i.e.,* the model is overconfident).

## Examples of Token-Level Compression

**<Token-Level Semantic Focus Analysis>**
Below are the key points where other agents **disagree** with your own reasoning.*
Concentrate on those disagreements and Keep or revise your answer accordingly.

<Copy the first agent message as the content is determined by internal attention that would be useful for current agent>
**One agent solution:** ```analysisWe need interpret reaction: 9-fluorenone + 2 equivalents of Selectfluor under photochemical? 400nm? Then product observed via 19F NMR with two fluorine signals: +29.8 ppm and -115.7 ppm. Which product matches? We need consider structure of products.

9-fluorenone has no fluorine. Fluorine signals appear from added fluorine atoms (from Selectfluor) presumably fluorination of the aromatic system. 19F NMR chemical shifts: +29.8 ppm is typical for aromatic fluorine attached to sp2 carbon in aromatic ring (like fluorobenzene ~7-8 ppm? wait fluorobenzene is 7 ppm. Actually 19F NMR for fluoride attached to benzene ring: about -113 to -115 ppm. For CF3 group ~ -65 ppm. For fluorines bound to carbonyl? Let's recall typical shifts: In 1,2-difluoro compounds, one signal can be upfield and one downfield depending on environment. For example, 2,4-difluoroaniline maybe at ~-110 and -45? Hard.

Let's analyze options:

(A) 1,5-difluoro-9H-fluoren-9-one. This would be fluorine at positions 1 and 5 of the fluorenone core. Aromatic fluorine signals for 1,5-difluoro; symmetrical? Two identical signals? But 19F NMR would show one signal (if equivalent)? Actually if two fluorines are meta to each other? Might be equivalent? 1,5-difluoro: two positions are equivalent if molecule symmetrical (flipped). For 1,5-difluoro-9-fluorenone, the two fluorines would be in symmetric positions. Would give a single 19F signal. But question says two fluorine signals, so (A) unlikely.

(C) 2,6-difluoro-9H-fluoren-9-one: also symmetrical? 2,6 are para on each phenyl ring? The two fluorines are in equivalent positions (mirror plane through center). Might again give single signal. So (C) unlikely.

(B) 2'-fluoro-[1,1'-biphenyl]-2-carbonyl fluoride. This is a different product: a biphenyl with fluorine on one ring and a carbonyl fluoride on other. This would have a single fluorine (the one at 2' position). Therefore only one 19F signal. So also unlikely.

(D) 4'-fluoro-[1,1'-biphenyl]-2-carbonyl fluoride. Also single fluorine.

...

 **One agent solution:** ```analysisWe need to analyze the described reaction: 9-fluorenone (structure: a fluorenone: a fused ring (fluorene) with a carbonyl at center? Actually 9-fluorenone is a 9-carbonyl in fluorene. 2 equivalents of Selectfluor used in acetonitrile, irradiated with 400 nm light. Reaction product has two fluorine signals: +29.8 ppm and -115.7 ppm in 19F NMR. Need to identify which structure among the options is correct product. Let's parse the options:

(A) 1,5-difluoro-9H-fluoren-9-one
(B) 2'-fluoro-[1,1'-biphenyl]-2-carbonyl fluoride

(C) 2,6-difluoro-9H-fluoren-9-one
(D) 4'-fluoro-[1,1'-biphenyl]-2-carbonyl fluoride

<Delele useless background information,>
We need to consider the reaction conditions. Selectfluor is used as a fluorination agent. Selectfluor is known to be a strong oxidant and fluorinating reagent, can generate radical cation. 9-Fluorenone has a central carbonyl. But 2 equivalents of Selectfluor might produce two fluorine atoms.

<Keep useful induction>
Looking at the product options: Both (B) and (D) are biphenyl derivatives with a carbonyl fluoride group at the 2-position of the biphenyl core. The difference is the position of a fluorine substituent on the biphenyl ring: 2' vs 4'.

<Cut out useless reasoning paths>
Options (A) and (C) are difluoro-substituted fluorene-9-one. 1,5-difluoro vs 2,6-difluoro arrangement on the fluorene core. Those are fluorinated at the aromatic positions adjacent to the carbonyl (perhaps positions that are 1 and 5, or 2 and 6 referencing fluorene numbering). But do we expect difluorination of the aromatic ring? Maybe the reaction could be electrophilic aromatic substitution with electrophilic fluorine; but Selectfluor is a fluorinating agent, providing F+. With 9-fluorenone, electrophilic aromatic substitution would likely occur at positions ortho/para to the carbonyl? etc) perhaps the substituents on the ring positions adjacent to the central carbonyl. I need to draw structure:
 The numbering scheme: positions 1, 3, 5, 7 correspond to the outer ring carbons; Let's quickly sketch: The central carbonyl is at position 9. I need to be precise. Let's step aside: We might not need exact numbering```

Using the reasoning from other agents as additional advice, can you give an updated answer? Examine your solution and that other agents step by step. Let's think step by step. Your last sentence should be 'The correct answer is (insert answer here).' e.g., The correct answer is (A).

Figure 21: Detailed example of token-level semantic focus. There is content from two agents in the figure. The first agent's answer was largely adopted, whereas the second agent's answer was largely cropped. <comments> shows places that were adaptively compressed by the model's attention mechanism.

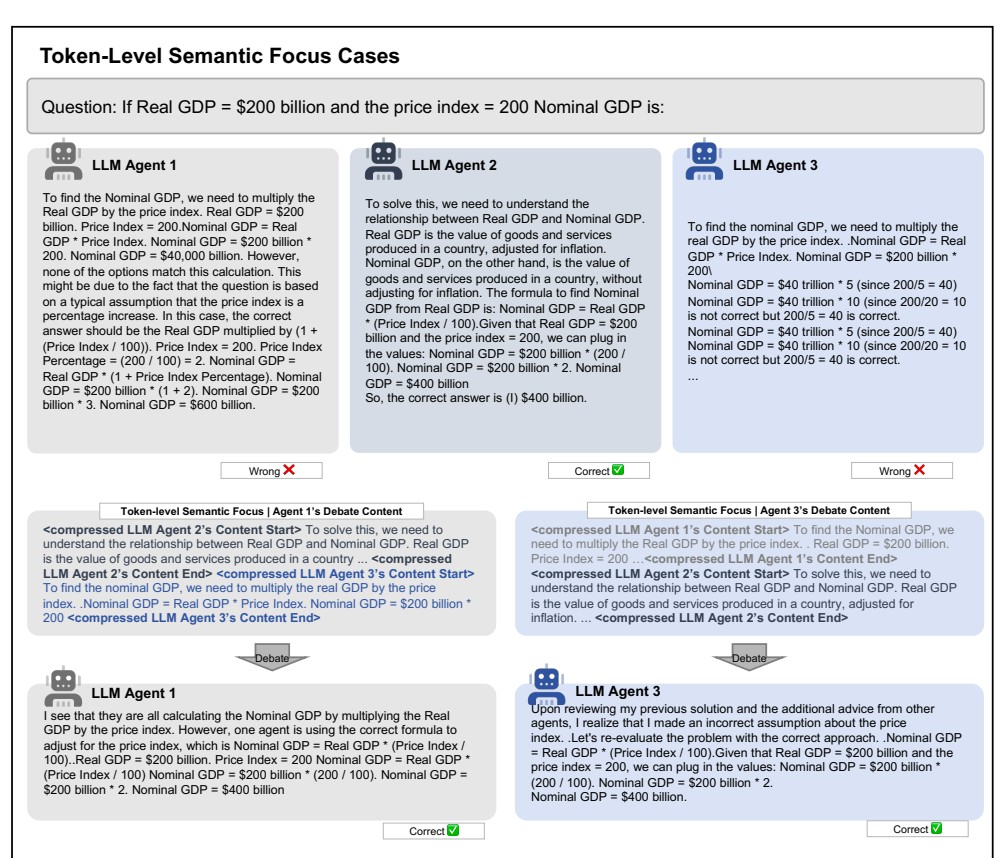

Figure 22: Visualization of token-level semantic focus helped agents correct their answers during the SID. Specifically, SID invokes a compressed debate round, highlighting disagreement-relevant spans across debate contents. Agent 1 and Agent 3 iteratively revise their reasoning based on focused inputs, ultimately correcting earlier errors and converging on the correct answer.

