# OpenReview forum: "SID: Multi-LLM Debate Driven by Self Signals"
_ICLR.cc/2026/Conference — ICLR 2026 Conference Withdrawn Submission_

### Official Review · Reviewer_mRDG · 2025-10-26

**Soundness:** 3
**Presentation:** 1
**Contribution:** 2
**Rating:** 4
**Confidence:** 3

**Summary:**

This paper proposed a multi-agent debate method that relies on internal signals such as token probabilities (or equivalently, entropy) as well as attention scores with respect to a predefined instruction prompt. It differs from most prior related works that only relies on external signals. Experiments show the effectiveness and efficiency of their proposed method.

**Strengths:**

The authors draw inspiration from the internal signals of LLMs and extend these ideas to the context of multi-agent debate, which, to the best of my knowledge, is a novel perspective.

They further conduct experiments on both LLMs and MLLMs to demonstrate the effectiveness and efficiency of the proposed method.

**Weaknesses:**

1. The writing of the paper is fairly problematic. For example, the format of citation (the use of \citep, \citet) leaves a bad first impression on the evaluation of the paper. Moreover, while the presentation is generally understandable, it lacks clarity and polish, which affects the readability and professionalism of the paper.

2. Intuitively, the proposed method seems primarily designed to improve efficiency. For example, the early-exit mechanism aims to avoid redundant generation when the model is sufficiently confident. However, it remains unclear why this mechanism, along with the attention-based compression approach, would lead to substantial performance improvements. The authors should provide deeper analysis or more convincing explanations to clarify how these efficiency-oriented components jointly enhance the multi-agent system's overall performance. This should be a key point of this paper.

3. Some closely related studies are missing. For example, [1] discusses the limitations and applicability of multi-agent debate systems. Including such references and comparing with them would strengthen the context and completeness of the paper.

[1] If multi-agent debate is the answer, what is the question, arXiv preprint arXiv:2502.08788

**Questions:**

See Weakness 2. The intuition why these efficiency-oriented methods can improve the overall performance is confusing. For example, if you do not use the early-exit mechanism (and only use the compression method), will you achieve a worse performance? If not, what are the potential reasons?

---

> ### Author Response · Authors · 2025-11-22
> **Response Part 1/2**
>
> Dear Reviewer mRDG,
> Thank you for your patience. Here's our response. We've also revised our manuscript to address your suggestions.
>
> ## W1: Citation format and writing.
>
> Thank you for pointing this out. We apologise for the confusion caused by the inconsistent use of the citation mark, as well as other stylistic issues. In the revised version, we have standardized all references to follow the ICLR author guidelines. We also carefully proofread the paper to improve clarity and readability. We are also very willing to discuss with you in the next stage to help us continuously improve the quality of our manuscript.
>
> ## W2 and Q1: Why “efficiency-oriented” components also improve performance?
>
> Thank you for raising this important point. At first glance, our early-exit and compression mechanisms indeed look primarily efficiency-driven. However, both are explicitly designed to reduce harmful debate and improve the signal-to-noise ratio in the context, which in turn improves accuracy.
> A well-known failure mode of multi-agent debate is that a correct initial answer can be “talked out of” during later rounds when exposed to noisy or incorrect reasoning from other agents. Our confidence-based early-exit gate is designed to alleviate this. Similarly, the compression mechanism is not random pruning for efficiency, but a way to focus the debate on disagreement-relevant evidence. Intuitively, this improves performance because later agents no longer need to attend to long, repetitive, and sometimes mutually reinforcing wrong reasoning. Instead, they see a concise, internally-derived “map” of where agents disagree and where high-information content lies.
>
> Our correction-flow analysis (Fig. 2(d) and the detailed plots in Appendix Figures 7–16) shows exactly this pattern: compared to vanilla MAD, SID reduces correct-to-wrong (C2W) transitions across rounds, while maintaining or increasing wrong-to-correct (W2C) transitions. This indicates that the early-exit gate is not merely shortening the process. It is selectively protecting high-confidence correct answers from being overturned by noisy debate.
>
> The ablation with and without semantic preservation (Table 3) supports this: when we remove semantic preservation and only keep raw token selection, accuracy drops sharply (from 46.83 to 34.50), showing that how we compress (disagreement-focused, sentence-complete spans) matters for correctness, not just efficiency.
>
> ## Direct Answer to What happens if we don’t use early-exit?
>
> Table 3 (MMLUpro, LLaMA3.1-8B) exactly addresses this:
>
> - **Baseline MAD**: 39.50 accuracy, token ratio 1.00
> - **Baseline MAD + Compression (no early-exit)**: 41.67 accuracy, token ratio 0.73
> - **Baseline MAD + Compression + Early Exit**: 46.83 accuracy, token ratio 0.53
> - **SID (full)**: 46.83 accuracy, token ratio 0.53
>
> From these numbers:
>
> - Compression *alone* already slightly improves performance over MAD (39.50 → 41.67) while saving tokens (1.00 → 0.73), consistent with the idea that focusing on disagreement-relevant spans helps both accuracy and efficiency.
> - Adding early-exit on top of compression yields a further substantial performance gain (41.67 → 46.83) *and* additional token savings (0.73 → 0.53).
>
> The intuitive reason is that without early-exit, some high-confidence correct answers are still forced into full debate, which occasionally corrupts them, whereas the early-exit gate prevents this over-debate. Together, the early-exit gate and attention-based compression are *not* merely cost-cutting tricks; they jointly (i) prevent unnecessary and harmful debate on already-correct cases, and (ii) sharpen the information presented in genuinely ambiguous cases. This is why we observe improvements in both efficiency *and* accuracy. We have revised our discussion part to explicitly state these findings.

---

> ### Author Response · Authors · 2025-11-22
> **Response Part 2/2**
>
> ## W3: Integrating discussion of related works.
>
> Thank you for pointing out this important work. We agree it is highly relevant and should be cited and discussed. Zhang et al. (2025) conduct a systematic evaluation of five representative MAD methods across nine benchmarks and four foundation models. They show that, in most settings, existing MAD frameworks do not reliably outperform strong single-agent baselines. We see our work as aligned with and complementary to these conclusions.
>
> Our results support this view: the baseline MAD in our setup offers only modest gains over single-round CoT, while consuming far more tokens. SID is explicitly designed as a response to this problem: it introduces a *selective* debate process driven by internal self-signals, rather than debating by default on every query.
>
> While Zhang et al. focus on model heterogeneity as a key lever for making MAD worthwhile, our work explores a different, orthogonal axis: self-signals–driven scheduling and compression. Instead of changing which models debate (heterogeneous vs homogeneous), we change when to debate (via confidence-based early-exit) and how much of the history to debate over (via attention-based compression). These ideas could, in principle, be combined with Heter-MAD.
>
> In the revised manuscript, we have integrated the paper into our related work to strengthen the context and completeness of the paper.

---

### Official Review · Reviewer_m9XW · 2025-10-28

**Soundness:** 2
**Presentation:** 3
**Contribution:** 2
**Rating:** 4
**Confidence:** 4

**Summary:**

The paper proposes Self-Signals–driven multi-LLM (SID) debate, a framework to leverage MAD with LLM internal signals: (1) model-level confidence, derived from token entropy; and (2) token-level semantic focus, from attention patterns conditioned on disagreement-oriented prompts. Experiments show that SID improves accuracy over MAD/DMAD while reducing token usage.

**Strengths:**

1. The paper presents another method to use self-signals for LLM multi-agent debate.
2. The author clearly described their method and it is easy to understand for the general reader.
3. The experiments show competitive performance.

**Weaknesses:**

1. The authors present this method as more token-efficient than other baselines, but we only see a comparison with MAD. How are the other methods comparing with it? How much of this efficiency is coming from the early-exit mechanism?
2. Selection of MAD/DMAD baselines seem reasonable, but fail to compare with a state-of-the-art LLM judge.
3. The authors fail to identify relevant related work in the are of LLM confidence/uncertainty in MAD research. And therefore, the use of self-signals cannot be claimed as novel. The authors make this claim in the first point of their contributions. They should rephrase as another method using self-signals in MAD. For instance, [ReConcile: Round-Table Conference Improves Reasoning via Consensus among Diverse LLMs](https://arxiv.org/abs/2309.13007) or [DebUnc: Improving Large Language Model Agent Communication With Uncertainty Metrics](https://arxiv.org/abs/2407.06426)
4. If I am not mistaken, the paper goes over the 9-page limit.

**Questions:**

1. How much of this efficiency is coming from the early-exit mechanism?
2. The authors present using self-signals in MAD as a novel technique, but as indicated in weaknesses other papers have already investigated the use of self-reported confidence and logit-based metrics as metrics for MAD. I'd suggest re-writing this contribution as another method of self uncertainty/confidence in MAD.
3. Given that the experiments are run on OOS models, why are confidence intervals not included?

---

> ### Author Response · Authors · 2025-11-22
> **Response Part 1/2**
>
> Dear Reviewer m9XW,
>
> Thank you very much for your careful reading and constructive feedback. We appreciate your time and address your concerns point by point below. Revisions corresponding to your comments will be incorporated into the updated manuscript.
>
> ## W1: Token efficiency, baselines, and contribution of early exit
>
> We appreciate the reviewer’s suggestion to include other baseline methods in the token efficiency comparison. We initially only compared with MAD as it is the raw baseline method without a multi-round prompting process (e.g., step-back prompting, which firstly prompts to obtain core ideas and secondly prompts to integrate the content to query the model). As DMAD utilizes these prompting methods, it is inherently more token-consuming than MAD and significantly more costly than our SID, even before considering early-exit.
>
> Based on DMAD’s official configurations (three agents × three rounds, with critique/reflection prompts), we summarize typical token ratios below:
>
> | Method | CoT | Self-Consistency | MAD | DMAD | SID |
> | --- | --- | --- | --- | --- | --- |
> | Token Ratio | 0.17 | 0.51 | 1.00  | 2.16 | 0.53 |
>
> As shown in the main results, SID achieves comparable or better accuracy than DMAD while using only about 25\% of its tokens.
>
> ## W2: Comparison with LLM-as-a-judge–style methods
>
> Thank you for raising the point about LLM-as-a-judge baselines.
>
> Our primary focus in this work is on MAD frameworks in the style of MAD/DMAD, where a fixed set of agents iteratively exchange messages under a shared protocol. Most prior MAD papers (e.g., Du et al., 2024; Liu et al., 2024) therefore compare against baselines such as CoT, Self-Consistency, MAD, and DMAD, but do not typically include full-fledged LLM-as-a-judge pipelines that invoke an additional large model to explicitly score or arbitrate each round. This is also why our initial baseline selection centered on MAD/DMAD-style methods: it follows the prevailing evaluation setup in the MAD literature and yields a controlled comparison under similar token budgets.
>
> In fact, several recent methods implicitly adopt a similar idea at the prompting level: for example, step-back or reflection prompts first ask the model to generate “core ideas” or self-critique summaries, and then use these as extra context in a second-stage query. These approaches, however, typically incur substantially higher token costs because they introduce additional calls and longer intermediate generations.
>
> To better address your concern, we additionally compare SID with Exchange-of-Thought (EoT), which is a multi-agent method that incorporates a consistency-based confidence signal. EoT derives a confidence score *post hoc* from the final discrete answers (e.g., options A/B/C/D) across rounds and agents by measuring how frequently the most common answer appears, and propagates this consistency signal to other agents during debate. We view EoT as a strong representative of LLM-as-a-judge–style consensus within the MAD family.
>
> We run EoT and SID under the same protocol on MMLU-Pro using LLaMA-3.1-8B-Instruct (three agents, two debate rounds), computing EoT’s confidence exactly as described in the original paper. The results are:
>
> | Method | CoT | MAD | EoT | SID |
> | --- | --- | --- | --- | --- |
> | Accuracy | 39.5 | 41.5 | 42.7 | **46.8** |
>
> EoT improves over standard MAD, confirming that consistency-based aggregation can indeed be beneficial. However, SID still achieves a clear additional gain, despite operating under the same multi-agent configuration.
>
> Du, Y., Li, S., Torralba, A., Tenenbaum, J. B. & Mordatch, I. Improving factuality and reasoning in language models through multiagent debate. in Proceedings of the 41st International Conference on Machine Learning vol. 235 11733–11763 ([JMLR.org](http://jmlr.org/), Vienna, Austria, 2024).
> Liu, Y., Cao, J., Li, Z., He, R. & Tan, T. Breaking Mental Set to Improve Reasoning through Diverse Multi-Agent Debate. in (2024).

---

> ### Author Response · Authors · 2025-11-22
> **Response Part 2/2**
>
> ## W3 and Q2: Related works on uncertainty in MAD
>
> Thank you for this very helpful clarification. We agree with the reviewer that the general idea of using uncertainty metrics is a shared high-level motivation across ReConcile, DebUnc, and our work. Both works clearly fall into the broader space of using model confidence/uncertainty signals in multi-agent settings. We will cite and discuss them explicitly in the revised related work. ReConcile uses confidence-weighted voting among diverse LLMs to reach consensus, where the confidence is from verbalizing between 0 to 1, and then is calibrated by a threshold. DebUnc computes the token-level probability for each agent and communicates this uncertainty to other agents. At the same time, we believe SID is more than “just another method using self-signals” because it uses internal LLM signals (logits and attention) in a structurally different way.
>
> - We propose a model-level early exit confidence mechanism to tell the system **when to debate**. ReConcile and DebUnc always run a full multi-round debate, but they do not change the *debate schedule* itself. In contrast, SID uses the model-l
> - evel confidence to make an early-exit decision. And we statistically demonstrate the effectiveness of the confidence in our Fig. 2 (c). Our design yields a training-free, model-agnostic gate that directly controls whether debate happens, and empirically leads to substantial token savings without hurting accuracy.
> - We further use a token-level compression mechanism to tell the system **how much history to debate over.** DebUnc’s attention scaling modulates attention weights across *entire* responses from different agents based on their scalar confidence, but it still feeds the full debate history to the model. SID instead uses self-attention as a *token-level semantic focus* signal: we inject a disagreement-oriented prompt, compute the maximum attention from prompt tokens to each token in other agents’ responses over all layers and heads, select a top-p subset, and then apply a SemanticPreserve procedure that merges and expands these tokens into coherent sentence-level spans. This produces a disagreement-focused, token-level summary driven entirely by internal attention.
> - SID is a unified framework that integrates the model-level and token-level mechanisms and leverages the self-signal in our debate algorithm. Extensive ablation experiments demonstrate that our design can both improve the accuracy and cut token usage by up to 30-40%.
>
> To avoid over-claiming and to better align with your comment, we will still soften and rephrase the first contribution, and avoid wording that suggests SID is the first to introduce self-signals into MAD.
>
> ## W4: Page limit
>
> Thank you for raising this concern. According to the ICLR 2026 author guidelines, the 9-page limit applies to the main text. There may be a misunderstanding about the optional Reproducibility Statement. In the author guideline, https://iclr.cc/Conferences/2026/AuthorGuide.
>
> It states that *Authors are strongly encouraged to include a paragraph-long Reproducibility Statement at the end of the main text before references to discuss the efforts that have been made to ensure reproducibility. ... This optional reproducibility statement is not part of the main text and therefore will not count toward the page limit.
>
> Therefore, our submission fits within the page limit.
>
> ## Q1: How much efficiency comes from early exit?
>
> We agree it is important to disentangle the contributions of compression and early-exit. Table 3 (LLaMA-3.1-8B on MMLU-Pro) already contains an ablation, and we will highlight it more clearly:
>
> - **Baseline MAD:** token ratio = **1.00**
> - **MAD + Compression only:** token ratio = **0.73**
> - **MAD + Compression + Early Exit (our SID):** token ratio = **0.53**
>
> From this, we see:
>
> - Compression alone reduces tokens by 27% (1.00 → 0.73).
> - Adding early exit on top of compression further reduces tokens by **20%** (0.73 → 0.53).
> - Overall, SID achieves a 47% reduction in token usage compared to vanilla MAD (1.00 → 0.53).
>
> Thus, early exit accounts for a substantial fraction of the total token savings (roughly 20 percentage points out of 47). We will explicitly state these numbers and emphasize that both components are necessary: compression reduces redundancy within each round, while early exit avoids unnecessary additional rounds by confident agents.
>
> ## Q3: Confidence intervals for OOS models.
>
> We appreciate this point about statistical rigor. To  address the reviewer’s concern and considering the computational cost, we additionally run three repeated evaluations on a representative MATH subset (Alg.) The observed variances remain small across all baselines:
>
> | Method | CoT | IO | MAD | DMAD | SID-v |
> | --- | --- | --- | --- | --- | --- |
> | Accuracy (std) | 61.8 (1.7) | 65.3 (2.0) | 61.9 (1.6) | 55.1 (1.5) | **67.2 (1.9)** |
>
> These results confirm that performance differences remain stable across repeated runs.

---

> ### Comment · Reviewer_m9XW · 2025-11-27
>
> Thank you for your response and updates of the paper.
>
> I believe some of these changes make the paper stronger and more faithful to its contributions. For instance, the ablation study now helps to better understand the effect of the added mechanisms to multi-agent debate. I have updated my score to reflect it
>
> I think the authors should still do a better analysis of the effects of the LLM-confidence. What happens when the LLMs are very certain about a wrong answer? How do we know the LLM confidence is correct?
>
> Also, some subfigures in Figure 2 should include a legend or a better description. For example, it isn't what the blue/gray line represents in Figures 2 (e), (f).

---

> > ### Author Response · Authors · 2025-11-28
> > **Thanks for your response for Reviewer m9XW 1/2**
> >
> > Dear Reviewer m9xW,
> >
> > Thank you very much for your follow-up comments and for updating your score. We are glad that the additional analyses made the paper clearer, and we appreciate your further suggestions on the LLM-confidence mechanism and Figure 2.
> >
> > ## Additional analysis of LLM confidence
> >
> > ### What happens when the LLM is very confident but wrong?
> >
> > We agree that over-confident errors are an important failure mode to analyze. In the revised manuscript, we have made this behavior explicit in Appendix E.
> >
> > - Figures 7–16 report, for each dataset and model, the distributions of our confidence metrics (NLL- and entropy-based) for correct (C) vs. wrong (W) predictions, together with a correction-flow analysis across rounds.
> > - Figures 17–20 then provide qualitative examples of early-exit cases, including overconfident wrong predictions, where the model’s internal confidence exceeds the early-exit threshold but the final answer is incorrect.
> >
> > From these examples, we observe that overconfident failures typically arise in cases where the intermediate reasoning is largely sensible (for example, correctly setting up definitions or intermediate steps) but the model makes a local slip (e.g., a final arithmetic or option-mapping error). Specifically, consider the example in Fig. 19 in the appendix, which asks to evaluate an expression involving $\lceil{\frac32}\rceil^2 + \lceil(\frac32)^2\rceil$. The correct value is $2^2 + 3 = 7$. In the trace shown in the figure, the agent correctly computes the intermediate quantities $(\frac32)^2 = \frac94$, $\lceil \frac94 \rceil = 3$ ,$\lceil \frac32 \rceil=2$ and all local algebraic steps are sensible.However, instead of returning the already correct value $\boxed{7}$, it goes on to apply the ceiling operator to an additional partial sum and then adds these values twice and finally outputs the over-confident but incorrect answer $9$.
> >
> > This illustrates a typical over-confident failure: the intermediate reasoning is mostly reasonable, but a small mistake in how intermediate results are combined leads to a confidently wrong final answer. This is consistent with the fact that our confidence is derived from token-level log-probabilities: if most of the reasoning trajectory is plausible, the aggregate NLL/entropy can still be low even when the final boxed answer is wrong.
> >
> > Crucially, our early-exit gate is calibrated so that such overconfident errors are relatively rare compared to the large set of high-confidence correct predictions:
> >
> > - In the confidence plots (Figures 7–16), the correct group consistently exhibits significantly lower NLL / entropy than the wrong group across multiple metrics, with statistical significance indicated in the figure captions.
> > - The correction-flow plots show that, under SID, we reduce C→W transitions while maintaining or increasing W→C transitions, compared to MAD (also summarized in Fig. 2(d) in the main text).
> >
> > In other words, while our confidence is not perfect and some “very confident but wrong” cases remain (as we explicitly showcase in Appendix E), the early-exit gate, on balance:
> >
> > 1. Protects many high-confidence correct answers from being overturned by noisy debate, and
> > 2. Does not introduce a large new class of confident errors.
> >
> > ### How do we know that the confidence estimate is reliable?
> >
> > Our analyses in Appendix E demonstrate that:
> >
> > - Across multiple datasets and models (Math subsets, MMStar, MMLU-Pro, GPQA), the distributions of confidence metrics for correct vs. wrong answers are systematically separated, and the differences are statistically significant in most settings (see Figures 7–16 and the associated p-value annotations).
> > - The correction-flow plots further show that the deltas of these confidence metrics across rounds align with beneficial corrections (W→C), while SID reduces harmful corrections (C→W) compared to MAD.
> >
> > These findings indicate that our confidence estimate, although imperfect, is well correlated with answer correctness and is therefore a suitable gating signal for deciding when to continue debate.
> >
> > This is also in line with recent studies on LLM uncertainty estimation, which show that token-level log-probabilities and entropy provide useful predictors of model error on QA and reasoning tasks, often outperforming external “model-as-a-judge” confidence signals, even though large models are not fully calibrated in the strict probabilistic sense (e.g., Kadavath et al., 2025; Tu et al., 2025; ). We have added a section in the Appendix. E to discuss the above questions.

---

> > ### Author Response · Authors · 2025-11-28
> > **Thanks for your response for Reviewer m9XW 2/2**
> >
> > ## Clarification of Figure 2
> >
> > Thank you for pointing out the ambiguity in the subfigures of Figure 2. In the revised version, we have added an explicit legend to Figures 2(e) , indicating that the red curve denotes accuracy and the gray curve denotes the token ratio (with the corresponding axes clearly labeled). We also explicitly describe the early-exit confidence threshold and the semantic-preservation ratio in the figure caption.
> >
> > ---
> >
> > We hope these clarifications address your concerns and make the role and behavior of the LLM-confidence mechanism more transparent. We are very grateful for your detailed feedback, which helped us improve both the analysis and the presentation.
> >
> > Reference
> >
> > Kadavath, S. et al. Language Models (Mostly) Know What They Know. Preprint at https://doi.org/10.48550/arXiv.2207.05221 (2022).
> >
> > Tu, W. *et al.* Ranked from Within: Ranking Large Multimodal Models Without Labels. in (2025).

---

### Official Review · Reviewer_MqxH · 2025-10-31

**Soundness:** 3
**Presentation:** 3
**Contribution:** 3
**Rating:** 6
**Confidence:** 3

**Summary:**

This paper proposes SID, a self-signals–driven framework for multi-LLM debate that replaces external controllers (e.g., LLM-as-a-judge) with two internal cues from the models themselves. The first cue is a model-level confidence score derived from token-probability statistics (entropy/NLL) that triggers an early-exit gate so confident agents stop debating; the second is a token-level “semantic focus” signal extracted from attention maps to compress debate histories into disagreement-relevant spans with a semantic-preservation heuristic. The early-exit mechanism can be implemented with a vocabulary-adaptive threshold or a lightweight calibrated classifier, with similar empirical performance and no training needed for the thresholded version. SID is evaluated on both LLMs and MLLMs, consistently improving accuracy over MAD/DMAD baselines while cutting token usage by up to ~30–40%.

**Strengths:**

* SID is well motivated for improving MAD from several critical perspectives
* The evaluation is comprehensive, covering a wide range of LLMs/MLLMs and different benchmarks
* The proposed components are novel, and be validated through fine-grained balation studies

**Weaknesses:**

* Missing baselines. There are several previous works with related confidence mechanisms, e.g., EoT [1] also incorporated a model-level confidence measurement. However, this work is not discussed or compared in this paper


[1] Exchange-of-Thought: Enhancing Large Language Model Capabilities through Cross-Model Communication

**Questions:**

* "Prompt-conditioned Attention Extraction" helps LLMs focus on the points of disagreement within a debate. Can we have the model do this by itself, rather than relying on external attention computations?

---

> ### Author Response · Authors · 2025-11-22
> **Response Part 1/1**
>
> Dear Reviewer MqxH,
>
> Thank you very much for your thoughtful and constructive review. We appreciate your time and helpful suggestions, and we have revised the manuscript accordingly (highlighted in the revised version). Below, we respond to your comments point by point.
>
> ## W1: Relation to EoT
>
> Thank you for indicating this paper. We will cite the paper and discuss it in the manuscript, as well as show it in the revised version. The discussion is as follows.
>
> EoT’s confidence is derived post hoc from the final discrete answers (e.g., option A/B/C/D) across rounds and/or agents, for example, by measuring how frequently the most common answer appears. This effectively treats consistency of verbalized outputs as a proxy for confidence. Such output-level reflection can be vulnerable to systematic errors: if a model (or group of models) consistently converges on an incorrect answer, the resulting “confidence” will still be high, even though the underlying reasoning is flawed. This risk is amplified when models have limited or unstable reflection ability; for instance, prior work such as Zhang et al. (2024) reports that reflection does not always improve accuracy, and smaller models in particular can suffer from degraded performance after reflection.
>
> Zhang, W. et al. Self-Contrast: Better Reflection Through Inconsistent Solving Perspectives. Preprint at https://doi.org/10.48550/arXiv.2401.02009 (2024).
>
> In contrast, SID’s model-level confidence is computed directly from internal self-signals (token-level logits/entropy / NLL) during the generative process, rather than from externally reflected answers. This design allows us to trigger early exit before producing and comparing multiple complete answers, thereby reducing redundant debate. Additionally, our model-level confidence mechanism is further complemented by a token-level self-signal (attention-based semantic focus) to compress debate history around disagreement-relevant spans.
>
> We further conduct the baseline comparison experiments with our method with Llama3.1-8B-Instruct on the MMLUpro dataset. The protocol is set as 3 agents, 2 debate rounds, with EoT confidence computed exactly the same as in the EoT paper. The result is summarized below:
>
> | Baseline  | CoT | MAD | EoT | SID |
> | --- | --- | --- | --- | --- |
> | Accuracy | 39.5 | 41.5 | 42.7 | 46.8 |
>
> It shows that EoT improves over standard MAD, validating that consistency-based aggregation can be beneficial. However, SID still achieves a clear improvement, confirming the effectiveness of internal self-signals. Additionally, our experiments (Fig. 2(c) in the manuscript) show that our model-level confidence has a statistically meaningful correlation with answer correctness, supporting its use as a more reliable gating mechanism than purely external reflection. That could be a potential reason our SID outperforms the EoT method. We will include this in our appendix.
>
> ## Q1: Can the model perform disagreement-focused extraction by itself?
>
> Thank you for this insightful question. In short, a purely prompt-based disagreement summary performs worse than our attention-based compression.
>
> We have indeed explored a purely prompt-based alternative where the model is asked to summarize disagreement on its own, without explicit attention computation. This variant is already included in our ablation study (Table 3), but we agree that it was not highlighted clearly enough. We will make this more explicit in the revised manuscript.
>
> Specifically, we add a baseline we refer to as “SID w/o Early Exit w/ Token-level Summary”. In this setting, after each debate round, we issue a summary prompt that instructs the model to: Identify the key points of disagreement in the debate history, and produce a concise summary of these disagreement-relevant spans, which is then used in place of our attention-based compressed context for the next round.
>
> Empirically, we observe that this prompt-based summary leads to a ~7.3% absolute accuracy drop compared to our attention-based SID variant. Also, it increases token usage by roughly 15%, due to the extra summarization calls and longer generated summaries.
>
> These results suggest that, at least in our setting:
>
> - Purely prompt-driven “self-summarization” of disagreements is less reliable than attention-based self-signals;
> - The model’s innate reflection and summarization ability is not yet strong enough to consistently preserve the fine-grained semantic cues needed for effective debate compression.
>
> By contrast, our prompt-conditioned attention extraction leverages internal attention patterns directly, providing a more stable and token-efficient approximation of where the model focuses semantically in the debate. We will explicitly emphasize this comparison and its implications in the revised ablation section and discussion.

---

### Official Review · Reviewer_65DL · 2025-11-01

**Soundness:** 2
**Presentation:** 3
**Contribution:** 2
**Rating:** 4
**Confidence:** 4

**Summary:**

Existing Multi‑Agent Debate (MAD) methods often rely on external structures and ignore self‑signals, which leads to redundant discussion and extra token consumption. The authors leverage entropy‑based model‑level confidence and token‑level semantic focus for early stopping and adaptive compression.

**Strengths:**

1. Experiments show the algorithm reduces token consumption (up to ~30%).
2. It also yields some performance improvement.
3. The paper is generally well written.
4. Although the use of logits for early stopping or attention to identify disputed segments can be found in the broad ML literature (see references below),  this paper is the first to apply them systematically within MAD.

References

Schuster, T., Fisch, A., Gupta, J., et al. Confident Adaptive Language Modeling. NeurIPS, 2022.

Laaouach, Y. HALT‑CoT: Model‑agnostic early stopping for chain‑of‑thought reasoning via answer entropy. Muslims in ML Workshop @ ICML 2025, 2025.

Yang, C., Si, Q., Duan, Y., et al. Dynamic Early Exit in Reasoning Models. arXiv:2504.15895, 2025.

Corallo, G., Papotti, P. FINCH: Prompt‑guided key‑value cache compression for large language models. Transactions of the Association for Computational Linguistics (TACL), 2024.

Fu, Q., Cho, M., Merth, T., et al. LazyLLM: Dynamic token pruning for efficient long context LLM inference. arXiv:2407.14057, 2024.

**Weaknesses:**

1. The core contribution is algorithmic, but the code is not open‑sourced.
2. The algorithm appears to constrain debate within homogeneous models, and calibrating confidence‑gating across different models seems challenging. If heterogeneous multi‑model debate is not supported, it underuses the potential of the MAD paradigm.
3. The evaluated benchmarks are somewhat limited (MMLU‑Pro, MATH, ScienceQA, MMStar); consider expanding to more benchmarks.
4.cBecause the method requires access to attention/logits, it is readily applicable only to open‑source/white‑box models; this is a limitation for MAD deployment scenarios.

**Questions:**

Zhang et al. report that, in many settings, Self‑Consistency (SC) can be both cheaper and more accurate than MAD. In your paper, however, SC seems unusually weak (e.g., GPT‑OSS‑20B on MMLU‑Pro: ~57). Could you explain why?

Zhang, H., Cui, Z., Wang, X., et al. If Multi‑Agent Debate Is the Answer, What Is the Question. arXiv:2502.08788, 2025.

---

> ### Author Response · Authors · 2025-11-22
> **Response Part 1/2**
>
> Dear Reviewer 65DL,
> Thank you for your patience. Here's our response. We've also revised our manuscript to address your suggestions.
>
> ## W1: Code availability
> Thank you very much for pointing this out. We now provide an anonymized repository containing our full implementation:
> https://anonymous.4open.science/r/SID-814D/. This repository is sufficient to reproduce the methods and experiments in our manuscript. We will also release the code publicly after the rebuttal.
>
> ## W2: Application to heterogeneous debate
>
> We acknowledge that we only used homogeneous models for our initial experiment settings due to time constraints in the current submission, following Du et al. (2024) and Liu et al. (2024) for a controlled comparison. We will make this explicit in Sec. 5.1 to avoid confusion.
>
> However, our method is algorithmically compatible with heterogeneous agents. Although our experiments are homogeneous, SID itself is not restricted to that case
>
> - The early-exit gate is computed per agent using the vocabulary-adaptive threshold (Sec 4.1), which depends only on that model’s own logits and vocabulary size, without requiring a shared calibration space across agents.
> - The compression mechanism operates entirely on each agent’s own attention maps over the shared debate context, and does not assume a particular architecture beyond the standard Transformer interface (logits + attention).
>
> Thus, mixing different models (e.g., one reasoning model plus two lighter general-purpose models) only requires access to their logits and attention, and SID’s mechanisms remain unchanged.
>
> To empirically validate this, we additionally run a heterogeneous debate study on MMLU-Pro with two base models: LLaMA-3.1-8B (model A) and GPT-OSS-20B (model B). We consider homogeneous pairs (A+A, B+B) and a heterogeneous pair (A+B), and report the accuracy of each agent’s final-round answer after debate:
>
> | Baseline | A+A, MAD | A+A, SID | B+B, MAD | B+B, SID | A+B, MAD (A/B) | A+B, SID (A/B) |
> | --- | --- | --- | --- | --- | --- | --- |
> | Accuracy | 41 | 47 | 65 | 71 | 65 / 65  | 64 / 72 |
>
> We observe the following:
>
> - Homogeneous settings.
>
>     SID consistently improves both models over MAD in homogeneous groups: A+A improves from 41 to 47, and B+B from 65 to 71. This mirrors the trends in our main experiments and confirms that SID’s early-exit and compression mechanisms remain beneficial regardless of the underlying model.
>
> - Heterogeneous setting (A+B).
>
>     In the A+B configuration, we report each agent’s own final prediction:
>
>     - Under MAD, both A and B converge to around 65% accuracy (65 / 65), suggesting that the weaker model A benefits from debating with the stronger model B, but the stronger model does **not** fully realize its potential within this unstructured debate.
>     - Under SID, the stronger model B reaches 72%, which is higher than any of the homogeneous baselines (including B+B with MAD at 65 and even B+B with SID at 71). Meanwhile, A remains competitive at 64%.
>
> Overall, these results demonstrate that SID not only extends naturally to heterogeneous multi-model debate but can also exploit model complementarity: it preserves the gains of multi-agent interaction for the weaker model while allowing the stronger model to achieve (and even slightly exceed) its best homogeneous performance.
>
> ## W3: Benchmark coverage
>
> Thank you for the suggestion to broaden the empirical evaluation.
> Our current experiments focus on MMLU-Pro, MATH, ScienceQA, and MMStar, which we selected to jointly cover:
>
> - General knowledge & reasoning: MMLU-Pro (multi-domain, more robust variant of MMLU).
> - Symbolic / structured reasoning: MATH, with diverse algebra/geometry/number-theory subsets.
> - Multimodal science reasoning: ScienceQA (text + image + lecture/hints).
> - Challenging vision–language understanding: MMStar.
>
> These four benchmarks jointly span a) text-only vs. multimodal input, b) factual vs. analytical vs. symbolic reasoning, and c) LLMs vs. MLLMs. We will clarify this motivation in Sec. 5.1. This design is consistent with recent works (Du et al., 2024; Liu et al., 2024).
>
> To further strengthen the coverage, we additionally include experiments on GPQA, a difficult benchmark targeting advanced science knowledge reasoning. Using GPT-OSS-20B under the same evaluation protocol and SID-v as our default setting, we obtain:
>
> | Method (GPQA) | CoT | IO | MAD | SID |
> | --- | --- | --- | --- | --- |
> | Accuracy (std) | 43.2 (1.7) | 41.4 (1.6) | 52.9 (2.0) | 54.8 (1.5) |
>
> SID consistently outperforms all baselines, demonstrating that our internal-signal-driven approach generalizes even to challenging scientific reasoning tasks. We will incorporate these results and discussions into the appendix.

---

> ### Author Response · Authors · 2025-11-22
> **Response Part 2/2**
>
> ## W4: Applicability for deployment scenarios.
>
> We appreciate this important clarification request. We agree that requiring access to logits and attention limits direct applicability to fully closed-source APIs. We have now made this limitation explicit in the Limitations section (Appendix B). At the same time, we would like to clarify the intended deployment scenarios:
>
> - SID is naturally suited to internal or on-premise deployments of proprietary models, where developers have access to internal signals. Many real-world MAD systems and tool-augmented agents fall into this category (e.g., internal inference servers rather than public APIs).
> - In these settings, SID can be used as an intermediate reasoning layer to prune redundant debate or compress history before forwarding results to an external API, thereby reducing total token usage and latency at the system level.
>
> Therefore, while SID is less suited for purely closed-source APIs, it remains practically useful in many real-world deployment settings and provides a general approach for improving the efficiency and performance of multi-agent debate systems.
>
> ## Q1: Why does Self-Consistency (SC) appear relatively weak?
>
> Thank you for pointing out this insightful paper. We agree that this is a critical study regarding the evaluation of MAD systems, and we will cite and discuss it in the revised version.
>
> In our experiments, SC is configured as follows (Sec. 5.1), in line with Liu et al. (2024): We use 3 SC samples, following prior work, and to keep the token budget comparable to MAD/SID. We apply the same base prompts as in CoT, and perform a majority vote over the sampled answers.
>
> Under this relatively small sampling budget, SC behaves as a moderately strong but not dominant baseline: For GPT-OSS-20B on MMLU-Pro, SC indeed improves over single-sample CoT and IO (69 vs. 61 and 64, respectively), consistent with Zhang et al.’s observation that SC can strengthen strong base models on challenging knowledge tasks. On hard math subsets (e.g., Algebra), CoT (85) outperforms SC (75), suggesting that in some cases repeated sampling reinforces a shared incorrect reasoning trajectory rather than correcting it.
>
> SC is most effective when the number of samples is high (e.g., 40 to 50), and the correct answer has non-trivial probability mass in the model’s output distribution. On our more difficult settings, the base model often exhibits “confident but wrong” behavior: once a flawed reasoning path is chosen, multiple SC samples gravitate toward similar incorrect chains. In Zhang et al.’s terminology, this corresponds to “consistently wrong” (WW) cases, where SC tends to converge to a systematic error rather than to the truth.
>
> We will explicitly cite Zhang et al. (2025) and add a short discussion in the revision to clarify this connection and the regimes where SC is relatively weaker in our experiments.
>
> ---
>
> Once again, we appreciate your careful reading and constructive suggestions. We believe these clarifications and revisions strengthen the paper both technically and empirically.
>
> Du, Y., Li, S., Torralba, A., Tenenbaum, J. B. & Mordatch, I. Improving factuality and reasoning in language models through multi-agent debate. in Proceedings of the 41st International Conference on Machine Learning vol. 235 11733–11763 ([JMLR.org](http://jmlr.org/), Vienna, Austria, 2024).
>
> Liu, Y., Cao, J., Li, Z., He, R. & Tan, T. Breaking Mental Set to Improve Reasoning through Diverse Multi-Agent Debate. in (2024).

---

> > ### Comment · Reviewer_65DL · 2025-11-26
> >
> > Since the authors have open-sourced their code, conducted additional experiments on heterogeneous models, and added more benchmark evaluations, I am inclined to raise my score.

---

> > > ### Author Response · Authors · 2025-11-26
> > > **Thanks for your response for Reviewer 65DL**
> > >
> > > We are glad that our responses have resolved your concerns, and we appreciate your positive remarks on the strength of our experiments and writing. Thank you for your valuable feedback, which will continue to guide and inspire our future work.

---

### Author Response · Authors · 2025-11-29
**Rebuttal Summary**

We would like to sincerely thank all reviewers, ACs and the committee for their time and efforts, especially given the unusual circumstances this year. We are very grateful that, during the discussion phase (before the score reset), **both reviewers who responded** indicated that the revised version strengthened the paper and **increased their scores**, so that the ratings at that time were **6, 6, 6, 4** ([a timestamped record is here](https://anonymous.4open.science/r/SID-814D/SID_rebuttal.pdf)). In particular:

- *Reviewer 65DL (Raised the score **from 4 to 6**):* “Since the authors have open-sourced their code, conducted additional experiments on heterogeneous models, and added more benchmark evaluations, **I am inclined to raise my score.**”
- *Reviewer m9XW (Raised the score **from 4 to 6**):* “I believe some of these changes make the paper stronger and more faithful to its contributions. For instance, the ablation study now helps to better understand the effect of the added mechanisms to multi-agent debate. **I have updated my score to reflect it.**”

To facilitate the review process, we summarize the main revisions (all highlighted in red in the updated manuscript)

- **Code release and reproducibility** (65DL)**.**

    We now provide an anonymized repository with the full implementation and scripts to reproduce all main experiments:

    `https://anonymous.4open.science/r/SID-814D/`.

- **Expanded benchmarks and heterogeneous debate** (65DL)**.** In addition to MMLU-Pro, MATH, ScienceQA, and MMStar, we added **GPQA** (advanced science reasoning) and new experiments with **heterogeneous agents** (LLaMA-3.1-8B + GPT-OSS-20B), showing that SID naturally extends to mixed-model debate and can even push the stronger model beyond its homogeneous performance.
- **Broader baselines, including LLM-as-a-judge–style methods** (m9XW, MqxH)**.** We added a comparison with **Exchange-of-Thought (EoT)** and a token-ratio analysis over CoT, Self-Consistency, MAD, DMAD, and SID. Under matched settings, SID outperforms EoT and achieves DMAD-level (or better) accuracy using only about 25% of DMAD’s tokens.
- **Expanded and clearer ablations** (m9XW, mRDG)**.** We disentangle the effects of MAD, attention-based compression, and early exit; and we compare our attention-guided compression with a prompt-only “disagreement summary”. Results show that early exit and self-signal–based compression jointly improve both efficiency and accuracy, while naive self-summary is less accurate and more expensive.
- **Model-confidence analysis** (m9XW)**.** A new appendix section analyzes our NLL/entropy-based confidence: distributions for correct vs. wrong answers, correction-flow statistics, and concrete over-confident failure cases. This shows that the signal is not an oracle but is strongly correlated with correctness and reduces harmful C→W transitions.
- **Positioning within related work** (all reviewers)**.** We explicitly discuss related works suggested by reviewers in our manuscript.
- **Writing, citations, and figures** (mRDG, m9XW). We fixed citation formatting, improved wording for clarity, and updated Figure 2 with legends and clearer captions.

We once again thank all reviewers for their constructive feedback, which has significantly improved the paper. We are very welcome for any further questions or comments.

---

### Note · Authors · 2026-01-29

I have read and agree with the venue's withdrawal policy on behalf of myself and my co-authors.

---

### Meta-Review · Area_Chair_8QkH · 2025-12-29

**Summary:**

The recommendation is primarily informed by the method's strict reliance on white-box access to model internals, specifically logits and attention maps, which severely limits its practical applicability. The technical contribution is viewed as incremental relative to existing literature on uncertainty-based early exits, while concerns persist regarding the calibration reliability of the internal confidence signals and the potential for overconfident errors to propagate through the system.

**Reviewer Concerns:**

The rebuttalresolved concerns regarding reproducibility and generalizability across model configurations. However, the method's fundamental dependence on white-box access constitutes an unresolved barrier, and the theoretical novelty of leveraging internal uncertainty signals remains incremental relative to existing literature.

**Reviewer Scores:**

Reviewer 65DL and m9XW would likely maintain their marginal scores as the provided empirical clarifications. Reviewer mRDG would probable lower their score, as lack of intuitive clarity remains insufficiently remediated.

---

### Decision · Program_Chairs · 2026-01-26

Reject